# Concept-level Multimodal Reasoning via Semantic Representation for Intent Recognition

## Abstract

Multimodal intent recognition is a fundamental task in understanding human communication, aiming to infer intent from heterogeneous modalities and serving as a cornerstone for developing human-centric systems. However, existing methods face two key challenges. First, they rely on entangled and modality-specific features, which hinder the derivation of interpretable representations across modalities. Second, they lack explicit reasoning mechanisms, making it difficult to capture high-level semantic dependencies and systematically link multimodal evidence to complex intents. To address these issues, we propose a novel method (ConMR) that conducts concept-level multimodal reasoning by jointly learning semantic concept representations and modeling concept relations. Specifically, we first leverage the Large Language Model (LLM) to generate high-quality intent-related concepts, providing explicit semantic anchors beyond shallow features. By supervising multimodal feature mapping through activation alignment, these concepts yield interpretable and discriminative representations. Building on this foundation, the concept-level multimodal reasoning module models concept-to-intent relations through LLM-guided relevance scores and infers inter-concept relations from activation patterns. By jointly exploiting these relations, it guides transparent reasoning paths from concepts to intents, thereby enhancing both accuracy and interpretability. Extensive experiments on two challenging datasets show that ConMR outperforms state-of-the-art methods with superior robustness and interpretability, laying a new paradigm for multimodal intent recognition.

## 1 Introduction

Multimodal intent recognition has emerged as a pivotal task in multimodal language analysis Zhang et al. (2022), aiming to interpret complex communicative intents by jointly modeling text, visual, and acoustic signals. This enables the capture of nuanced semantics in context-rich interactions and is essential for various applications in intelligent transportation Creß et al. (2023), chatbots Fan et al. (2022), medical diagnosis Tiwari et al. (2022), and other interactive systems Paul et al. (2022); Mi et al. (2019). Research in this field is pioneered by MIntRec Zhang et al. (2022), the first benchmark dataset that formally defines the task and establishes initial baselines by adapting multimodal fusion methods from sentiment analysis Hazarika et al. (2020); Rahman et al. (2020); Tsai et al. (2019). Building on this foundation, MIntRec2.0 Zhang et al. (2024a) expands dataset scale and label taxonomy, providing a more comprehensive benchmark and stimulating growing research interest.

Existing multimodal intent recognition methods focus on exploiting diverse feature cues to enhance intent understanding. A prominent line of research incorporates supervision from intent labels to capture fine-grained interactions and emphasize intent-relevant signals, such as TCL-MAP Zhou et al. (2024) and MVCL-DAF Hu et al. (2025). Another direction emphasizes cross-modal alignment strategies, where SDIF-DA Huang et al. (2024) incrementally aligns audio and video with text through a shallow-to-deep framework to enhance semantic consistency before fusion. Beyond such fusion strategies, several works address task-specific challenges by refining feature representations. For instance, CAGC Sun et al. (2024) seeks to improve discriminability in ambiguous contexts, Inmu-Net Zhu et al. (2024) aims to filter redundancy and preserve task-relevant semantics, and DuoDN Chen et al. (2024) utilizes causal modeling to improve interpretability and reduce spurious correlations. More recently, MIntOOD Zhang et al. (2024b) aims to extend multimodal intent recognition to open-world settings for both in-distribution classification and out-of-distribution de-

tection, while LGSRR Zhou et al. (2025) represents the first attempt to leverage Large Language Models (LLMs) to guide multimodal intent recognition, which still suffers from coarse-grained intermediate semantics and a reasoning process limited to logical relations. Despite these significant advancements, multimodal intent recognition still face two significant challenges. First, existing methods predominantly operate at the feature level, relying on entangled and abstract representations that leave a substantial gap between low-level multimodal signals and the nuanced semantics of human intent. Second, they lack explicit and structured multimodal reasoning mechanisms capable of modeling the interplay between high-level semantic representations, which makes it difficult to construct transparent and discriminative paths that bridge raw inputs to complex intents.

To address these issues, we propose ConMR, a novel framework that elevates multimodal reasoning to the concept level by jointly learning semantic concept representations and modeling inter-concept relations, thereby enabling interpretable and structured reasoning across modalities. The motivation comes from concept bottleneck models (CBMs), which demonstrate that explicit concept supervision enhances interpretability by bridging raw features and task labels. However, CBMs are typically confined to single-modal settings and neglect inter-concept relationships, limiting their effectiveness in complex multimodal scenarios. Building on these insights, ConMR introduces two complementary modules. The Concept Representation Learning module leverages Gemini-2.5 to generate and filter high-quality intent-related concepts, ensuring semantic coverage and task relevance via discriminability and coverage scores Yang et al. (2023b). To ground multimodal signals in these concepts, we compute activation scores by measuring their similarity to concept embeddings using modality-specific pretrained models, which then supervise transformation matrices that project heterogeneous features into a unified concept space to yield interpretable and discriminative concept representations. The Concept-level Multimodal Reasoning module builds upon these representations by modeling both concept-to-intent and inter-concept relations. For concept-to-intent reasoning, intent-conditioned relevance scores generated by Gemini-2.5 are leveraged to guide a weighting network through MSE loss to selectively reinforce concept features, while inter-concept reasoning exploits activation patterns across concepts to modulate final logits and refine intent predictions. By jointly leveraging these two relations, ConMR establishes transparent reasoning paths from multimodal inputs to intents, enhancing both interpretability and performance.

Our contributions are summarized as follows: (1) We propose ConMR, the first method that pioneers concept-level multimodal reasoning by transforming entangled multimodal features into explict concepts and reasoning over their relations, enabling robust intent understanding. (2) Experiments on challenging benchmarks highlights that ConMR achieves significant gains over state-of-the-art methods while offering faithful concepts and transparent reasoning pathways for enhanced interpretability. (3) By tackling the challenge of constructing anchors for complex multimodal semantics, our concept-level paradigm demonstrates strong generalizability and extensibility, providing a principled foundation for reliable and scalable multimodal reasoning across diverse tasks.

## 2 RELATED WORKS

### 2.1 MULTIMODAL INTENT RECOGNITION

Multimodal intent recognition aims to understand user intents by integrating verbal and non-verbal signals from real-world interactions. To facilitate progress in this field, MIntRec Zhang et al. (2022) introduces the first benchmark dataset, covering a wide range of interaction scenarios and offering baseline models via multimodal fusion strategies. It is later extended by MIntRec2.0 Zhang et al. (2024a), which expands the data scale and intent categories, offering a more comprehensive and challenging benchmark. Based on these datasets, early methods on multimodal intent recognition mainly center on effective fusion strategies. For example, MulT Tsai et al. (2019) first applies cross-modal transformers for modality alignment, MAG-BERT Rahman et al. (2020) introduces adaptive attention gating to integrate acoustic and visual cues with text, and MISA Hazarika et al. (2020) further learns modality-specific and modality-invariant features jointly via multiple optimization objectives. Subsequent studies begin to explore richer inter-modal relations and more efficient aggregation. DuoDN Chen et al. (2024) disentangles semantic- and modality-oriented representations through dual networks with counterfactual intervention, whereas InMu-Net Zhu et al. (2024) employs an information bottleneck to suppress redundancy and retain intent-relevant signals. Recently, contrastive learning has become a central paradigm for reinforcing critical semantics, with TCL-

MAP Zhou et al. (2024) employing token-level alignment for intent semantics in real multimodal settings and CAGC Sun et al. (2024) capturing cross-video contextual dependencies. To further enhance fusion by addressing modality imbalance, MVCL-DAF Hu et al. (2025) leverages multi-view contrastive objectives, and SDIF-DA Huang et al. (2024) designs a shallow-to-deep multimodal fusion framework. Besides, MIntOOD Zhang et al. (2024b) adopts a multi-task learning framework for intent recognition and out-of-scope detection, thereby enhancing performance and robustness.

## 2.2 CONCEPT BOTTLENECK MODELS

Concept Bottleneck Models (CBMs) Koh et al. (2020) aims to improve interpretability by introducing an intermediate layer of human-understandable concepts before final predictions, where each neuron corresponds to a semantically meaningful concept. However, their reliance on manually annotated concept labels limits scalability, particularly when supervision is costly or infeasible. To overcome this, recent studies start to explore methods that automatically discover concepts without human annotations. For instance, Label-Free CBM Oikarinen et al. (2023) leverage pretrained vision-language models like CLIP Radford et al. (2021) to extract semantic concepts and align bottleneck neurons via similarity-based matching, thereby avoiding the need for labeled concept data. Following this direction, several studies Yang et al. (2023a); Srivastava et al. (2024) further advance automatic concept generation and concept selection, aiming to filter irrelevant concepts and enhance semantic alignment for improved interpretability and scalability. Beyond automatic concept annotation, CBMs are further extended for greater flexibility. Post-hoc and open-vocabulary approaches, such as PCBM Yuksekgonul et al. (2022), OpenCBM Tan et al. (2024), and ALBM Zhang et al. (2025), decouple concept discovery from training by retrofitting frozen backbones into concept bottlenecks, allowing users to add, remove, or refine concepts during inference without gradient updates or extra labels. To enhance interpretability and reliability, probabilistic and energy-based extensions such as ProbCBM Kim et al. (2023), ECBM Xu et al. (2024), SCBM Vandenhirtz et al. (2024), and EQ-CBM Kim et al. (2024) model concept activations as distributions or energy functions, yielding calibrated uncertainty estimates and principled interventions that capture concept dependencies. In parallel, interactive and continual learning frameworks like CooP Chauhan et al. (2023), CBM-HNMU Xiong et al. (2025), and language-guided CBMs Yu et al. (2025) incorporate human feedback under uncertainty to refine representations and dynamically expand concept libraries for evolving tasks. Although CBM research has made notable strides in interpretability, it remains confined to surface-level contribution scores of concepts rather than capturing their intrinsic semantics, which fundamentally constrains its performance.

## 3 METHOD

In this section, we present ConMR, a novel method that advances multimodal reasoning from entangled feature-level processing to a structured paradigm of concept-level inference, as illustrated in Figure 1. It consists of two key components, the Concept Representation Learning module described in Section 3.1 and the Concept-level Multimodal Reasoning module detailed in Section 3.2.

## 3.1 CONCEPT REPRESENTATION LEARNING

To provide explicit and reliable semantic anchors for multimodal reasoning, our approach bridges low-level inputs and high-level intent semantics via a structured concept space that ensures both interpretability and discriminative power. Inspired by Oikarinen et al. (2023), we first employ LLMs to generate diverse concepts related to intents for each modality, and then project each sample onto these anchors to obtain concept-level representations. This design grounds the framework in meaningful cues while injecting structured signals that form a robust basis for downstream reasoning.

**Concept Generation.** To ensure the generality of our method, the concept generation process relies solely on task descriptions and intent labels, without requiring any real data. Formally, given the intent set $\mathcal{I} = [y_1, y_2, \ldots, y_m]$, we leverage the advanced analytical capabilities of Gemini-2.5 Flash to generate modality-specific concepts for each intent label. The resulting concepts are aggregated into three sets $\{C^T, C^V, C^A\}$, corresponding to text, video, and audio, respectively. To evaluate the quality of these candidates, we focus on two key properties including their semantic relevance to the target intent and redundancy with other concepts within each modality. Specifically, utilizing

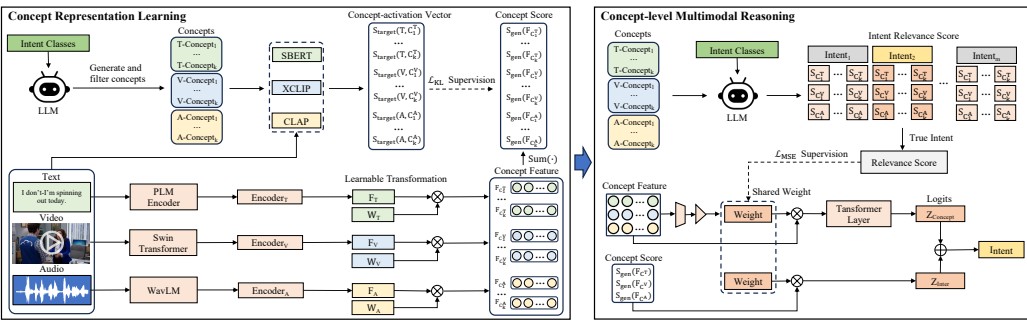

Figure 1: Overall architecture of ConMR with two main modules. The Concept Representation Learning module employs an LLM to generate and filter intent-related concepts, then projects text, video, and audio features into the concept space, supervised by activation scores from SBERT, XCLIP, and CLAP to yield interpretable representations. The Concept-level Multimodal Reasoning module builds on these representations by modeling concept-to-intent relevance, where intent-conditioned scores guide selective amplification of salient concepts, and inter-concept dependencies, where activation dynamics refine logits through semantic interactions.

the text embedding model Li & Li (2024), we obtain vector representations $\mathcal{E}(c)$ for each concept $c$ and $\mathcal{E}(y)$ for its corresponding intent $y$. We then compute the cosine similarity between the concept and the intent $\cos(\mathcal{E}(c), \mathcal{E}(y))$ to assess relevance, and the pairwise similarity between concepts $\cos(\mathcal{E}(c), \mathcal{E}(c'))$ to assess redundancy. By applying a threshold-based filtering strategy on these similarities, we obtain modality-specific concept sets that are diverse, intent-relevant, and of high quality. To further derive the final concept set, we follow LaBo Yang et al. (2023b) and apply submodular selection to each modality concept set, jointly maximizing discriminativeness and coverage, where $C^M = \{c_1^M, \dots, c_k^M\}$ denotes the concept set associated with modality $M \in \{T, V, A\}$.

Following Oikarinen & Weng (2023) Building on these high-quality modality-specific concepts, we obtain their activation patterns on training samples by employing modality-specific encoders SBERT Li & Li (2024), XCLIP Ma et al. (2022), and CLAP Wu* et al. (2023) to compute the association scores $S_{\text{tgt}}(T, c_i^T)$ between each concept and the raw multimodal input $X = \{T, V, A\}$. These scores are aggregated into a concept-activation vector, which subsequently serves as the supervision signal to guide the mapping from multimodal features to concept representations

$$S_{\text{tgt}} = \left[ S_{\text{tgt}}(T, C^T), S_{\text{tgt}}(V, C^V), S_{\text{tgt}}(A, C^A) \right]. \tag{1}$$

Further details including step-by-step descriptions and prompts, are provided in Appendix B.

**Feature-to-Concept Transformation.** Following Zhang et al. (2022), we first derive modality-specific token sequences from raw inputs using pretrained backbones, and then refine them with Transformer Vaswani et al. (2017) encoders to enhance contextual expressiveness before pooling to global features. For the textual modality, the widely-used pretrained BERT Devlin et al. (2019) embeds the input utterance into token representations $E^T = \{t_1, t_2, \dots, t_{n_T}\} \in \mathbb{R}^{n_T \times d}$. For the visual modality, video frames are processed by the strong backbone ResNet He et al. (2016) to obtain spatial-temporal embeddings $E^V = \{v_1, v_2, \dots, v_{n_V}\} \in \mathbb{R}^{n_V \times d}$. For the audio modality, the audio waveform is encoded by Wav2Vec Baevski et al. (2020) into $E^A = \{a_1, a_2, \dots, a_{n_A}\} \in \mathbb{R}^{n_A \times d}$. Here, $n_T$, $n_V$, and $n_A$ denote the sequence lengths of each modality, and $d$ is the shared embedding dimension. To further strengthen representation capacity and capture rich contextual dependencies, each modality's token sequence is processed by a modality-specific encoder, formalized as:

$$\hat{T} = \text{Enc}_T(E^T), \quad \hat{V} = \text{Enc}_V(E^V), \quad \hat{A} = \text{Enc}_A(E^A), \tag{2}$$

where $\text{Enc}_M(\cdot)$ comprises Transformer Vaswani et al. (2017) layers to enhance semantic coherence. Finally, we apply mean pooling over the refined token sequences to obtain compact global features

$$F_T = \text{MeanPool}(\hat{E^T}), \quad F_V = \text{MeanPool}(\hat{E^V}), \quad F_A = \text{MeanPool}(\hat{E^A}). \tag{3}$$

Subsequently, we construct an interpretable concept space by aligning opaque features with semantic anchors, enabling downstream reasoning to operate at the concept level. Inspired by CLIP-Dissect Oikarinen & Weng (2023) and label-free Concept Bottleneck Models Oikarinen et al.

(2023), we use target activations $S_{\text{tgt}}$ to supervise a learnable transformation matrix $W$, which maps modality-specific features onto these activation profiles. Unlike prior work that remains at the activation score level without recovering explicit concept representations, ConMR extracts disentangled concept vectors directly from the features. Formally, the transformation matrix is composed in a modality-aware manner as $W = [W_T; W_V; W_A]$, where each $W_M \in \mathbb{R}^{k \times d}$ for $M \in T, V, A$ corresponds to a specific modality, $k$ denotes the number of concepts, and $d$ is the feature dimensionality. The $i$-th row of $W^M$ encodes the activation weights of the $i$-th concept across the feature dimensions of modality $M$. Disentangled concept vectors are obtained via element-wise multiplication

$$F_{c^T} = F_T \cdot W_T, \quad F_{c^V} = F_V \cdot W_V, \quad F_{c^A} = F_A \cdot W_A, \tag{4}$$

yielding concept representations across different modalities. The reliability of this disentanglement process is ensured through supervision from the target activations $S_{\text{tgt}}$. Since the resulting concept vectors inherently encode activation information, we derive the concept-level activation scores $S_{\text{gen}}$ by directly applying a summation function over all dimensions of each concept vector, accordinig to CLIP-Dissect. The fidelity of the activation pattern is enforced by minimizing the Kullback–Leibler (KL) divergence between the generated activation distribution $S_{\text{gen}}$ and the target distribution $S_{\text{tgt}}$ obtained from the original multimodal inputs and textual concepts

$$\mathcal{L}_{\text{KL}} = \sum \text{softmax}(S_{\text{tgt}}) \log \frac{\text{softmax}(S_{\text{tgt}})}{\text{softmax}(S_{\text{gen}})}. \tag{5}$$

Minimizing this divergence guarantees semantic consistency of concept activations, establishing a reliable basis for multimodal reasoning. Further explanations are provided in Appendix A.

### 3.2 Concept-level Multimodal Reasoning

Based on semantically grounded representations, we propose a concept-level multimodal reasoning paradigm that integrates concept-to-intent relations and inter-concept relations within a coherent framework. The concept-to-intent relation is guided by LLMs to generate intent-conditioned relevance scores, ensuring precise semantic alignment between individual concepts and intent categories. In parallel, the inter-concept relation leverages concept activation scores to capture interactions among co-activated concepts, allowing higher-order semantics to naturally emerge. This design enables nuanced understanding of complex intents by explicitly modeling how atomic concepts correlate with intents and how they interact with each other to form composite semantics.

**Concept-to-Intent Relation.** To establish reliable supervision, we leverage the semantic reasoning capability of LLMs to generate intent relevance scores. Given the modality-specific concept sets $C^T, C^V, C^A$ and the intent category set $\mathcal{I}$, the LLM evaluates the semantic relatedness between each concept and each intent, producing a relevance score within the range $[-1, 1]$. The absolute value of the score indicates the degree of relatedness, while the sign specifies whether the semantic direction of the concept is consistent or contradictory with the target intent. Since these scores are intent-conditioned, during training we use the scores $S_{c^T}$, $S_{c^V}$, and $S_{c^A}$ corresponding to the ground-truth intent to supervise the weight generation. Concretely, given the concept representations $F_{c^T}$, $F_{c^V}$, and $F_{c^A}$, a shared weight generator composed of a multi-layer perceptron (MLP) is applied to produce intent-conditioned relevance weights. Formally, the weight matrix is obtained as

$$W = \tanh(\text{MLP}([F_{c^T}; F_{c^V}; F_{c^A}])), \tag{6}$$

where $[\cdot; \cdot]$ denotes vector concatenation and the $\tanh(\cdot)$ activation guarantees that the generated weights fall within the range $[-1, 1]$, consistent with the LLM-derived intent relevance scores. During training, $W$ is optimized under MSE supervision with $S_{c^T}$, $S_{c^V}$, and $S_{c^A}$, ensuring semantically grounded reweighting of the concept features, which is defined as

$$\mathcal{L}_{\text{MSE}} = \frac{1}{3k} \sum_{i=1}^{k} \left( \left| W[i] - S_{c_i^T} \right|^2 + \left| W[i+k] - S_{c_i^V} \right|^2 + \left| W[i+2k] - S_{c_i^A} \right|^2 \right). \tag{7}$$

The reweighted concept features are obtained as $F_c = [F_{c^T}; F_{c^V}; F_{c^A}] \cdot W$ and subsequently passed through the Transformer Vaswani et al. (2017) layers to refine semantic interactions, before being projected through a classification head to yield the concept-to-intent logits $Z_{\text{concept}}$.

**Inter-Concept Relation.** While concept-to-intent relevance highlights the most salient concepts for a given intent, intent understanding often emerges from interactions among multiple concepts. To capture such dependencies, we leverage the activation scores $S_{\text{gen}}$, which quantify the degree of activation of each concept. These scores are further modulated by the intent-conditioned weights $W$, ensuring that the modeled interactions are aligned with the target intent. The resulting interaction-aware representations are then aggregated and projected into the intent space to produce the inter-concept logits. Through this process, $Z_{\text{inter}}$ encodes higher-order semantic relations by amplifying the co-activation patterns of intent-relevant concepts, thereby providing complementary evidence to the direct concept-to-intent alignment captured by $Z_{\text{concept}}$.

Finally, the two reasoning pathways are seamlessly integrated to produce the unified intent logits

$$Z = Z_{\text{concept}} + Z_{\text{inter}}. \tag{8}$$

The final intent distribution is obtained via the softmax function applied over $Z$, and optimized with the standard cross-entropy loss

$$\mathcal{L}_{\text{CLS}} = -\frac{1}{B} \sum_{i=1}^{B} \log \frac{\exp(Z_{y_i})}{\sum_{j=1}^{m} \exp(Z_j)}, \tag{9}$$

where $B$ is the batch size, $Z_{y_i}$ is the logit corresponding to the label $y_i$ of the $i$-th sample. Finally, the overall training objective combines three components of KL divergence loss for concept activation alignment, MSE loss for weight supervision, and cross-entropy loss for classification

$$\mathcal{L} = \alpha\mathcal{L}_{\text{KL}} + \beta\mathcal{L}_{\text{MSE}} + \gamma\mathcal{L}_{\text{CLS}}, \tag{10}$$

where $\alpha$, $\beta$, and $\gamma$ are hyperparameters that balance the relative contributions of each loss.

## 4 EXPERIMENTS

**Datasets.** To evaluate our method, we conduct experiments on two challenging multimodal benchmarks. MIntRec Zhang et al. (2022) is a fine-grained multimodal intent recognition dataset comprising 2,224 high-quality samples across text, video, and audio modalities, annotated with 20 intent categories. We adopt the official split, with 1,334 samples for training, 445 for validation, and 445 for testing. MIntRec2.0 Zhang et al. (2024a) extends MIntRec by significantly enlarging the dataset scale and expanding the intent label space to 30 categories, while maintaining the same multimodal setting. Importantly, the two datasets are non-overlapping, and in our experiments we use the in-domain portion of MIntRec2.0, which follows the official partition of 6,165 training samples, 1,106 validation samples, and 2,033 test samples.

**Baselines.** Following Zhang et al. (2022), we adopt state-of-the-art methods as baselines: (1) MulT Tsai et al. (2019) employs directional cross-modal attention to capture inter-modal interactions without strict alignment; (2) MISA Hazarika et al. (2020) decomposes features into modality-specific and modality-invariant subspaces, then applies self-attention for efficient fusion; (3) MAG-BERT Rahman et al. (2020) introduces an adaptive gating mechanism that refines text embeddings using offsets derived from nonverbal modalities; (4) MMIM Han et al. (2021) learns mutual information maximization across modalities to preserve complementary information and enhance cross-modal consistency; (5) TCL-MAP Zhou et al. (2024) exploits token-level contrastive learning to strengthen textual representations by integrating visual and acoustic cues, improving semantic acquisition and multimodal integration; (6) SDIF-DA Huang et al. (2024) leverages a shallow-to-deep interaction module to align and fuse modalities across different semantic granularities; (7) MIntOOD Zhang et al. (2024b) adopts a weighted feature fusion network to better capture multimodal representations under multiple optimization objectives; (8) MVCL-DAF Hu et al. (2025) applies multi-view contrastive learning to exploit diverse perspectives for robust multimodal alignment and fusion; (9) LGSRR Zhou et al. (2025) leverages LLM to derive structured intermediate semantic descriptions, upon which it conducts logic-driven relational reasoning; (10) Gemini-2.5 Flash is a leading MLLM with strong multimodal reasoning abilities, evaluated in zero-shot, few-shot, and CoT settings. The CoT design directs a structured reasoning path involving global perception, concept identification, relational analysis, and intent inference.

**Evaluation Metrics.** Following Zhang et al. (2024a), we assess model performance using standard metrics including accuracy (ACC), F1-score (F1), precision (P), recall (R), weighted F1-score (WF1), and weighted precision (WP), where higher scores indicate better performance.

Table 1: Main results comparing ConMR with baselines on the MIntRec and MIntRec2.0 datasets.

| Methods | MIntRec | | | | | | MIntRec2.0 | | | | | |
|---|---|---|---|---|---|---|---|---|---|---|---|---|
| | ACC (↑) | F1 (↑) | P (↑) | R (↑) | WF1 (↑) | WP (↑) | ACC (↑) | F1 (↑) | P (↑) | R (↑) | WF1 (↑) | WP (↑) |
| MulT | 71.50 | 68.29 | 69.37 | 68.20 | 71.25 | 71.75 | 60.17 | 52.71 | 58.35 | 52.47 | 58.92 | 60.09 |
| MISA | 71.37 | 68.23 | 69.02 | 68.45 | 71.43 | 72.33 | 57.49 | 51.60 | 54.69 | 50.87 | 57.09 | 58.09 |
| MAG-BERT | 71.32 | 66.94 | 67.39 | 67.63 | 70.81 | 71.32 | 60.34 | 54.58 | **58.63** | 54.34 | 59.51 | 60.42 |
| MMIM | 71.37 | 68.09 | 69.10 | 68.41 | 71.25 | 72.23 | 55.81 | 49.81 | 53.40 | 49.42 | 55.13 | 56.51 |
| SDIF-DA | 71.10 | 68.32 | 70.36 | 67.43 | 70.89 | 71.55 | 58.42 | 52.07 | 54.08 | 51.87 | 57.63 | 58.00 |
| TCL-MAP | 72.99 | 68.74 | 68.48 | 69.89 | 72.63 | 72.95 | 58.52 | 53.00 | 55.18 | 52.91 | 57.80 | 58.25 |
| MIntOOD | 72.81 | 69.13 | 69.93 | 69.27 | 72.56 | 72.97 | 58.23 | 51.38 | 55.64 | 51.36 | 56.98 | 58.16 |
| MVCL-DAF | 73.84 | 71.03 | 72.45 | 71.26 | 73.53 | 74.55 | 58.96 | 53.79 | 54.81 | 53.88 | 58.51 | 58.86 |
| LGSRR | 73.26 | 70.77 | 72.08 | 70.07 | 72.97 | 73.15 | 60.30 | 55.04 | 57.57 | 54.17 | 59.77 | 60.20 |
| Gemini-2.5 | 55.12 | 52.46 | 56.03 | 54.19 | 55.74 | 56.30 | 40.93 | 41.35 | 42.67 | 39.82 | 42.48 | 46.09 |
| *+Few-shot* | 61.44 | 59.91 | 60.04 | 56.77 | 60.18 | 62.52 | 43.15 | 42.27 | 42.06 | 40.53 | 45.78 | 47.81 |
| *+CoT* | 66.52 | 63.03 | 65.52 | 64.88 | 67.74 | 65.45 | 44.19 | 42.25 | 44.58 | 44.39 | 44.33 | 48.78 |
| ConMR | **75.91** | **73.06** | **73.48** | **73.37** | **75.76** | **76.20** | **60.89** | **55.46** | 58.05 | **55.22** | **60.04** | **60.46** |

**Implementation Details.** For concept scoring, we use SBERT Li & Li (2024), X-CLIP Ma et al. (2022), and CLAP Wu* et al. (2023), with pretrained checkpoints from Hugging Face, and adopt the submodular selection method from LaBo Yang et al. (2023b) to obtain high-quality subsets of LLM-generated concepts. For training MIntRec and MIntRec2.0, we apply Adam Kingma & Ba (2015) with gradient clipping, early stopping, and a customized learning rate scheduler, along with an improved Pareto-inspired optimizer Sener & Koltun (2019) to dynamically balance task losses. The text encoder is initialized with BERT-large-uncased, and all hidden sizes are set to 1024. We train with a batch size of 16, setting the concept set size to 100 for MIntRec and 150 for MIntRec2.0, and using learning rates of 2e-5 and 1.5e-6, respectively. For fair comparison, results are averaged over five runs with seeds 0-4, and all experiments are conducted on an NVIDIA Tesla V100-SXM2 GPU. Additional details are provided in the Appendix C.

**Results.** The performance of ConMR and strong baselines on both datasets is presented in Table 1, with the best results highlighted in bold and the second-best results underlined. On the MIntRec dataset, ConMR demonstrates clear superiority, achieving average improvements of more than 3% over most methods and thereby showcasing its strong discriminative capacity. Even when compared with the strongest competitor MVCL-DAF, ConMR preserves consistent gains from 1.03% to 2.22% across all metrics, underscoring its stable and robust advantage. On the more challenging MIntRec2.0 dataset, which features a larger scale and more diverse intent categories, ConMR continues to demonstrate state-of-the-art performance. Although MAG-BERT achieves comparable results in P metric, ConMR surpasses all baselines on the remaining metrics, striking a stronger overall balance. Notably, ConMR delivers the largest gains in R, surpassing the strongest baseline by 0.88% and demonstrating its ability to capture subtle semantic cues. Compared with LGSRR which also leverages LLM capabilities, ConMR consistently outperforms across all metrics due to its finer-grained concept representations and more comprehensive concept-level reasoning. Furthermore, despite Gemini-2.5's strong multimodal reasoning capabilities, it struggles with complex intent semantics, trailing ConMR by over 10% in both zero-shot and few-shot settings. Even with detailed CoT providing a clear concept-based reasoning path, Gemini-2.5 still lags by over 8%, clearly indicating that LLMs remain limited in performing effective reasoning for intent comprehension. These results collectively validate the effectiveness of our concept representation and multimodal reasoning strategy, demonstrating improved robustness and performance in intent understanding.

## 5 DISCUSSION

### 5.1 ABLATION STUDIES

To further asses the contribution of each component in ConMR, we perform ablation studies on both datasets, with results summarized in Table 2. In the Concept Representation Learning module,

Table 2: Ablation studies on the MIntRec and MIntRec2.0 datasets.

| Ablations | MIntRec | | | | | | MIntRec2.0 | | | | | |
| --- | --- | --- | --- | --- | --- | --- | --- | --- | --- | --- | --- | --- |
| | ACC (↑) | F1 (↑) | P (↑) | R (↑) | WF1 (↑) | WP (↑) | ACC (↑) | F1 (↑) | P (↑) | R (↑) | WF1 (↑) | WP (↑) |
| w/o $\mathcal{L}_{KL}$ | 74.61 | 71.97 | 72.55 | 72.25 | 74.50 | 75.19 | 60.28 | 54.91 | 57.35 | 54.71 | 59.40 | 59.81 |
| w/o $W$ | 73.75 | 70.18 | 72.08 | 69.85 | 73.41 | 74.17 | 51.62 | 44.93 | 47.45 | 44.98 | 50.80 | 51.92 |
| w/o $\mathcal{L}_{MSE}$ | 74.07 | 70.78 | 71.44 | 71.69 | 73.56 | 74.15 | 60.10 | 54.80 | 56.92 | 54.73 | 59.25 | 59.52 |
| w/o $Z_{concept}$ | 36.36 | 23.06 | 38.83 | 23.36 | 30.40 | 40.26 | 58.27 | 46.99 | 52.07 | 46.36 | 55.77 | 56.19 |
| w/o $Z_{inter}$ | 74.52 | 71.88 | 72.77 | 72.02 | 74.51 | 75.38 | 60.85 | 54.90 | 57.29 | 54.75 | 59.82 | 60.06 |
| Full | **75.91** | **73.06** | **73.48** | **73.37** | **75.76** | **76.20** | **60.89** | **55.46** | **58.05** | **55.22** | **60.04** | **60.46** |

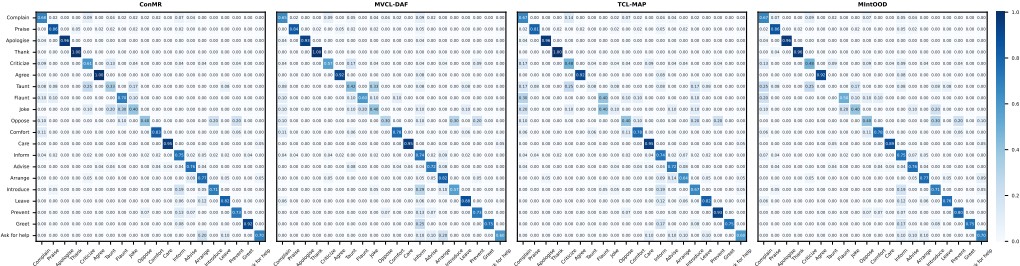

Figure 2: Confusion matrices of ConMR and three strong baselines on the MIntRec dataset.

removing the KL divergence loss (w/o $\mathcal{L}_{KL}$) causes performance drops of over 1% on MIntRec and over 0.5% on MIntRec2.0 across most metrics, underscoring the importance of reliable concept activations for building robust representations and enabling precise intent understanding. Besides, a severe degradation is observed when the learnable transformation $W$ is replaced with linear layers with activations (w/o $W$), with metrics on MIntRec2.0 dropping by more than 7%, which highlights the critical role in generating robust concept representations. In the concept-level multimodal reasoning module, removing $\mathcal{L}_{MSE}$ (w/o $\mathcal{L}_{MSE}$) results in performance drops from 0.49% to 2.28% across all metrics on both datasets, confirming the importance of LLM-based intent relevance score supervision. Furthermore, ablating the concept-to-intent pathway (w/o $Z_{concept}$) causes a severe collapse, with accuracy on MIntRec dropping to 36.36% and F1 on MIntRec2.0 falling to 46.99%. This setting aligns with concept bottleneck models by relying solely on activation scores, highlights the superiority and robustness of our concept-to-intent pathway in handling diverse scenarios. Meanwhile, removing the inter-concept relation pathway (w/o $Z_{inter}$) leads to moderate but consistent declines, demonstrating that modeling concept interactions provides complementary benefits for composing higher-order semantics and achieving more comprehensive intent understanding. Additional ablations on concept selection are provided in Appendix D.

## 5.2 ANALYSIS OF CLASSIFICATION PERFORMANCE

To gain deeper insights into classification performance, we visualize confusion matrices of ConMR and three best-performing baselines on the MIntRec dataset in Figure 2. From an overall perspective, ConMR achieves the highest classification accuracy in 16 out of 20 intent categories, clearly surpassing the three baselines, which only obtain the best results in 6, 4, and 8 categories respectively. This broader coverage demonstrates that ConMR utilizes concept-level reasoning to disentangle subtle semantic differences and maintain balanced performance across fine-grained intents, enabling it to excel in challenging categories with overlapping meanings. Specifically, ConMR not only achieves perfect predictions on relatively simple intents such as *Thank* and *Agree*, but also outperforms the second-best baseline by more than 10% on more complex categories such as *Flaunt*. These gains highlight the strength of our reasoning mechanism in enhancing both discrimination robustness across diverse scenarios. Despite its limitations on *Joke*, ConMR still achieves the highest score, demonstrating strong generalization. The difficulty highlights the need for future models to integrate external knowledge to better interpret such complex and nuanced cases.

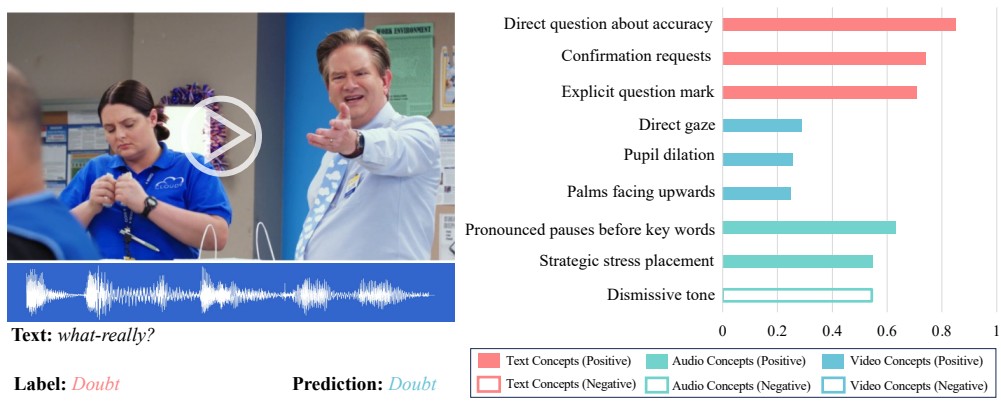

Figure 3: Case study illustrating top concepts leveraged by ConMR in multimodal reasoning.

## 5.3 CASE STUDY

To demonstrate the effectiveness and interpretability of ConMR, Figure 3 presents a representative example from the MIntRec2.0 test set, displaying the top three concepts from each modality by relevance score to transparently illustrate how our method grounds reasoning in meaningful multimodal cues and delivers faithful explanations. In this example, textual cues like *direct questions about accuracy* and *confirmation requests* decisively support the intent with high positive relevance scores by directly aligning with linguistic uncertainty. Besides, visual cues such as *direct gaze* and *pupil dilation* reinforce the intent by capturing non-verbal expressions and gestures, complementing the textual evidence. In the audio modality, rhythmic cues such as *pronounced pauses* and *strategic stress placement* strengthen intent recognition by highlighting vocal emphasis, while the negatively correlated *dismissive tone* reduces confusion with conflicting intents like *criticize* or *taunt*. By explicitly modeling both supportive and opposing concept-intent associations, ConMR achieves interpretable and robust multimodal reasoning that amplifies complementary cues while suppressing misleading signals. More cases on both datasets are presented in Appendix F.

## 5.4 PERFORMANCE WITH DIFFERENT LLM BACKBONES

In this experiment, we investigate the generalizability of the proposed concept-level multimodal reasoning framework across different large language model (LLM) backbones. To this end, the default Gemini-2.5 for concept generation and intent relevance scoring is replaced with three widely adopted alternatives, GPT-5, Gemini-2.0 and Qwen3-8B. All models are trained and evaluated under identical settings on the MIntRec dataset, with results summarized in Table 3. It is evident that ConMR delivers consistently strong results across all LLM backbones, underscoring its adaptability to different semantic priors. Notably, Gemini-2.5 achieves the best overall performance, while other backbones also yield comparable results, clearly outperform-

Table 3: Performance of ConMR with different LLM backbones on the MIntRec datastet.

| Backbones | ACC (↑) | F1 (↑) | P (↑) | R (↑) | WF1 (↑) | WP (↑) |
|---|---|---|---|---|---|---|
| GPT-5 | 75.33 | 72.23 | 73.33 | 72.32 | 75.00 | 75.46 |
| Gemini-2.0 | 75.19 | 72.52 | 73.15 | 72.87 | 75.54 | 75.19 |
| Qwen3-8B | 75.15 | 72.79 | **74.41** | 72.56 | 75.03 | 75.90 |
| Gemini-2.5 | **75.91** | **73.06** | 73.48 | **73.37** | **75.76** | **76.20** |

ing baseline methods on most metrics. Specifically, when GPT-5 and Gemini-2.0 serve as backbones, their performance deviates from the original ConMR by just over 1% on a single metric. Remarkably, Qwen3-8B not only sustains consistently high overall performance but even surpasses the original ConMR on the P metric, further demonstrating ConMR's strong generalization in low-resource settings and highlighting its practical utility. This stable performance across diverse backbones demonstrates that ConMR does not rely on a specific LLM, but instead establishes a flexible and robust paradigm for multimodal understanding. At the same time, the observed differences among backbones highlight the benefits of leveraging more advanced models, which provide richer concept generation and more reliable intent relevance estimation. These findings jointly position ConMR as a potentially general foundation for multimodal reasoning tasks.

## 6 Conclusion

This paper presents ConMR, a pioneering method that transitions multimodal reasoning from opaque feature-level patterns to the explict concept-level paradigm, offering stronger performance and more trustworthy reasoning pathways. ConMR first grounds multimodal features in LLM-generated and semantically filtered concepts by learning transformation matrices under activation-based supervision, thus constructing a unified and discriminative concept representation space. Building on this foundation, it leverages concept-to-intent relations through relevance weighting and models inter-concept interactions via activation patterns, enabling structured and interpretable multimodal reasoning. Experiments on two benchmarks demonstrate consistent gains over state-of-the-art baselines in both performance and interpretability, while evaluations with different LLM backbones further validate the robustness and generalizability of ConMR. This adaptability highlights the promise of the concept-level paradigm as a principled foundation for advancing multimodal reasoning across diverse tasks. The limitations of this work and the details regarding the use of LLMs are provided in Appendix H and Appendix I.

## 7 Ethics statement

This work adheres to the ICLR Code of Ethics, which emphasizes research integrity, transparency, and responsibility. First, we respect privacy by using only publicly available datasets under open-source licenses, ensuring that our work benefits society and human well-being. Second, we commit to honesty and transparency by faithfully reporting our methods and limitations, and by ensuring reproducibility. Thirdly, our proposed algorithm has the potential to generate positive societal impacts, such as enhancing human–computer interaction, supporting mental health assessment, and improving customer service automation. While any technology can be applied for both beneficial and harmful purposes, we believe that the ethical deployment of our method ultimately depends on the safeguards implemented and the specific context of use.

## 8 Reproducibility Statement

We are committed to promoting reproducibility and advancing research in this field. To this end, we will release the full source code of ConMR upon publication, and an anonymized example code is provided in the supplementary materials. All implementation details are explicitly described in the main text and appendix: (1) the feature extraction models are introduced in Section 3.1, (2) the evaluation datasets and implementation details are provided in Section 4, (3) the exact steps and prompts used for LLM-based concept generation and intent relevance scoring are included in Appendix B, (4) the full hyperparameters and training settings are provided in Appendix C. These efforts ensure that our work can be faithfully reproduced and serve as a foundation for future research.

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

## A  FURTHER EXPLANATIONS OF FEATURE-TO-CONCEPT TRANSFORMATION

**Label-Free Concept Bottleneck Model.** The Label-Free Concept Bottleneck Model (CBM) Oikarinen et al. (2023) addresses two central limitations of conventional CBMs, including the reliance on labeled concept annotations and the significant performance drop compared to standard neural networks. Instead of requiring manually curated concepts, Label-Free CBM leverages large foundation models such as CLIP Ma et al. (2022) and GPT-3 to automatically generate and filter a set of candidate concepts. It then learns a projection from the backbone feature space to a concept bottleneck layer by optimizing the alignment between projected neurons and CLIP-derived concept activations. Finally, a sparse linear classifier is trained on top of the bottleneck layer, enabling predictions to be expressed as transparent linear combinations of human-understandable concepts. This pipeline enables Label-Free CBM to scale to large datasets, train efficiently within hours, and operate with minimal human intervention. Empirically, Label-Free CBM retains accuracy close to the original backbone while providing faithful concept-based explanations of model decisions.

**CLIP-Dissect.** CLIP-Dissect Oikarinen & Weng (2023) is a method that automatically generates human-interpretable descriptions for individual neurons in visual networks without requiring manual annotations. Leveraging the pretrained CLIP model, given a concept set $C$ and a probing dataset $\mathcal{D}$, the method constructs a concept activation matrix $P$ that records the cosine similarity between each element in $\mathcal{D}$ and each concept in $C$. The dataset $\mathcal{D}$ is then passed through the visual network under analysis, and the activation pattern of a neuron $k$ is collected as a vector $q_k$. Based on a similarity function $\text{sim}(c_i, q_k, P)$, the concept associated with neuron $k$ is determined by computing $\arg\max_i \text{sim}(c_i, q_k, P)$. This approach enables systematic analysis of neuron-level representations without manual supervision, and demonstrates the feasibility of obtaining textual concepts aligned with the visual feature space in a manner without human annotation.

**Our Method.** Inspired by these works, we adopt a CLIP-like architecture to transform multimodal features into concepts enriched with explicit semantics. However, our objective differs substantially from theirs. Rather than treating these scores as the final output, we additionally seek to obtain explicit feature representations of concepts in the feature space of our modality-specific encoder. In other words, we aim to learn a mapping from each concept to a feature vector that preserves the cosine similarity with inputs defined by the CLIP-like models, thereby grounding concepts in stable and semantically meaningful representations. Concretely, we treat the CLIP-like architecture as a teacher model and distill its supervision into a trainable concept matrix where each row encodes a distinct concept and each column spans the dimensions of the feature space. This matrix can be viewed as a trivial "text encoder" that maps each concept to a feature vector. Training proceeds by minimizing the discrepancy between the cosine similarities produced by the teacher and those derived from the student matrix. As a result, each concept is assigned a stable and semantically meaningful embedding in the feature space of the modality-specific encoder. These distilled concept-level feature representations provide a reliable foundation for subsequent concept-level inference.

# B  ADDITIONAL DETAILS FOR CONCEPT GENERATION

In this section, we describe the implementation pipeline for generating concepts in concept representation learning and assigning intent relevance scores in concept-level multimodal reasoning. The overall pipeline is outlined step by step in Appendix B.1, followed by detailed explanations of key components, including the prompt for concept generation of Step1 in Appendix B.2, the submodular selection strategy of Step5 in Appendix B.3, and the prompt for intent relevance score assignment of Step6 Appendix B.4.

## B.1  CONCEPT PIPELINE

**Step1 Generate Concepts.** We do not ask domain experts to manually propose concepts, since this process can easily miss important perspectives, bring in subjective bias, and require additional cost. Instead, we use Large Language Models (LLMs) to automatically generate candidate concepts. For each intent $i$ and modality $j$, we prompt Gemini-2.5 Flash through its API to act as an intent recognition expert and produce a candidate set $C^{i,j}$ that can identify intent $i$ from modality $j$ alone. The exact prompts and the rationale behind the design are described in Appendix B.2.

**Step2 Remove concepts close to intents.** We use a text embedding model to represent each concept $c$ and its intent $y$ as vectors $\mathcal{E}(c)$ and $\mathcal{E}(y)$. We then compute their cosine similarity $cos(\mathcal{E}(c), \mathcal{E}(y))$ and remove all concepts with a similarity greater than 0.8. This prevents trivial concepts from entering the pool and helps improve both interpretability and the reliability of CLIP based supervision.

**Step3 Remove concepts close to each other.** Using the embeddings from Step 2, we calculate cosine similarity between every pair of concepts $cos(\mathcal{E}(c), \mathcal{E}(c'))$. For any pair with similarity above 0.9, we discard the concept ranked earlier. This step ensures that the final concept pool does not contain redundant concepts. It also saves computation, provides broader coverage in Concept Level Reasoning, and improves reasoning quality.

**Step4 Embed inputs and concepts.** After filtering, we use the CLIP like encoder of each modality to embed both the input and its concepts. For a sample input $\mathcal{X}_j$, with a text encoder $\mathcal{T}$ and a modality encoder $\mathcal{M}$ trained together, we obtain $\mathcal{T}(\mathcal{X}_j) \in \mathbb{R}^h$ and $\mathcal{M}(c_i) \in \mathbb{R}^h$ for all $c_i \in C_j$. With these embeddings, we can directly run submodular selection and score computation, which reduces extra inference cost.

**Step5 Submodular selection.** Following LaBo Yang et al. (2023b), we apply submodular selection to the subsets obtained in Step 3 and Step 4. The selection is guided by a submodular function that balances discriminability and coverage. Using a greedy algorithm, we select a fixed number of concepts that achieve a good tradeoff between the two. This balance is important. If the subset is too discriminative, it may miss useful variations. If it is too redundant, it reduces interpretability and efficiency. More details on the objective and our implementation can be found in Appendix B.3. After selection, with the refined concept set, we assign concept annotations to each input by computing cosine similarity between inputs and concepts, which produces compact but semantically meaningful supervision.

**Step6 Intent relevance scoring.** To incorporate concept–to-intent relations into multimodal reasoning, we again rely on LLM to perform the annotation. For the final concept set $C_{Final}^{i,j}$ associated with intent $i$ and modality $j$, we provide all concepts to the LLM and ask it to assign a score between $-1$ and $1$ to each concept, reflecting its contribution to predicting the target intent. Carefully designed prompts guide the model to produce complete, consistent, and reliable scores, while keeping the output in natural language form. After generation, we parse the raw output with regular expressions to extract and format the scores, and verify that the number of scores matches the number of concepts. If the counts do not align, we request the LLM to regenerate the output until a valid sequence is obtained. The exact prompts and the rationale behind the design are described in Appendix B.4.

## B.2 Prompts for Concepts Generation

We illustrate our concept-generation prompt with the text modality under the intent *Complain*, where content within {} is intent-specific and content within [] is modality-specific. In this prompt design, we explicitly introduce the task setting of multimodal intent recognition to the LLM, thereby anchoring its generation within the target domain. To ensure that the generated outputs serve as effective concept candidates, we provide detailed requirements and constraints, enforcing that the resulting concepts are sufficiently **numerous**(1), **semantically accurate**(2, 5, 6, 10), **comprehensive in coverage**(3, 4, 7), **interpretable**(5, 6, 10), **unimodal**(8), and **observable**(8). In addition, we include illustrative examples of concepts to exploit the LLM's few-shot learning capability, encouraging it to produce more coherent and contextually grounded concepts. Through these design choices, we improve the quality of concept candidates at the generation stage itself, ensuring that the resulting concept bank is both high-quality and reliable for downstream reasoning.

```
Your goal is to generate some concepts that describe a person's [text
    features] when they are trying to '{Complain}', which will help
    in understanding their intentions. Follow the rules below
    strictly:
1. List 20-50 features.
2. Output only the features. Do not include any introductions,
    numbers, or explanations.
3. Cover a wide range of [linguistic aspects]: vocabulary, syntax,
    pragmatics, discourse, etc.
4. Ensure features are independent and non-redundant.
5. Use short, precise sentence segments.
6. Use direct, simple English words.
7. Ensure the features are unique to '{Complain}' and not general or
    shared with other intentions.
8. Only describe language features that can be obtained from [speech
    text], [not audio, visual, or emotional cues].
9. Include the most important aspects relevant to '{Complain}'.
10. Keep descriptions concise but complete.
11. Below are 2 examples to guide you. You can include them in your
    output:
Examples for '{Complain}':
{
    "Negative words"
    "Repetitive complaints"
}
```

### B.3 SUBMODULAR SELECTION STRATEGY

In a single modality, given a concept set $S_y$ associated with a class $y$, the task of selecting an optimal subset $C_y$ with desirable properties can be formulated as a submodular optimization problem. Specifically, for a set function $\mathcal{F} : 2^{S_y} \to \mathbb{R}$ that is monotone and satisfies the diminishing returns property, a greedy algorithm Nemhauser et al. (1978) provides a $(1 - 1/e)$-approximation to the subset of fixed size that maximizes $\mathcal{F}$.

Building on this framework, LaBo Yang et al. (2023b) defines a submodular function $\mathcal{F}$ that balances *discriminability* and *coverage* in concept selection:

$$\mathcal{F}(C_y) = \alpha \cdot \sum_{c \in C_y} D(c) + \beta \cdot \sum_{c_1 \in S_y} \max_{c_2 \in C_y} \phi(c_1, c_2), \tag{11}$$

where $D(c)$ measures the discriminability of a concept $c$, and $\phi(c_1, c_2)$ quantifies the coverage between two concepts.

**Discriminability Score.** For a concept $c$ and class $y$, the similarity is defined as

$$Sim(y, c) = \frac{1}{|\chi_y|} \sum_{x \in \chi_y} x \cdot \mathcal{T}(c), \tag{12}$$

where $\chi_y$ denotes the set of samples labeled as $y$, and $\mathcal{T}$ is the text encoder. This similarity is normalized as

$$\overline{Sim(y'|c)} = \frac{Sim(y', c)}{\sum_{y'' \in Y} Sim(y'', c)}, \tag{13}$$

and the discriminability score is then defined as

$$D(c) = \sum_{y' \in Y} \overline{Sim(y'|c)} \cdot \log\left(\overline{Sim(y'|c)}\right). \tag{14}$$

Maximizing $D(c)$ ensures that selected concepts are highly associated with only a few classes, thereby enhancing discriminability.

**Coverage Score.** Coverage is measured by the cosine similarity between concepts:

$$\phi(c_1, c_2) = \cos\left(\mathcal{T}(c_1), \mathcal{T}(c_2)\right). \tag{15}$$

Minimizing $\phi$ encourages the selection of semantically diverse concepts, thus improving coverage.

**Our Implementation.** In our framework, we adopt a CLIP-like architecture, which provides both a text encoder $\mathcal{T}$ and a modality-specific encoder $\mathcal{M}$. To compute discriminability scores, we directly reuse the normalized representations of $x$ and $\mathcal{T}(c)$ obtained during the concept scoring process. We employ the apricot library to perform submodular selection, consistent with LaBo, and set $\alpha = 1$ and $\beta = 0.5$. For each class, we select 5 concepts as the final subset. Following the instructions reported in the LaBo GitHub issues, we also modified the apricot source code to ensure correct functionality.

### B.4 PROMPTS FOR INTENT RELEVANCE SCORING

We illustrate our prompt for relevance scoring under the intent *Complain*, where the content within {} is intent-specific. In this prompt, we instruct the LLM to act as a human expert in intent recognition. The prompt explicitly states that concept–to-intent relations can be either positive or negative, and emphasizes the need to consider co-occurrence patterns among concepts in order to obtain more rational, comprehensive, and accurate scores. Regarding the output format, the model is asked to return both the index and the original text of each concept, ensuring that all concepts are properly annotated. We further require the model to output scores in a structured manner that can be easily extracted using regular expressions. In addition, the model is asked to provide a brief explanation after assigning scores, which encourages more stable and well-justified outputs. Through analysis of the scoring results, we observe that the LLM consistently assigns strongly positive scores to concepts generated under the correct intent, suggesting that the model is indeed capable of producing reasonable scores.

Table 4: Key hyperparameters for MIntRec and MIntRec2.0 experiments.

| | MIntRec | MIntRec2.0 |
|---|---|---|
| Epochs / Warmup | 100 / 10 | 100 / 8 |
| EarlyStopping patience | 3 | 4 |
| Batch sizes(train/val/test) | 16/8/8 | 16/8/8 |
| Concept capacity ($t/a/v$) | 100 / 100 / 100 | 150 / 150 / 150 |
| Layers ($t/v/a$) | 2 / 2 / 2 | 2 / 3 / 4 |
| Concept layers | 4 | 4 |
| Attention heads | 8 | 8 |
| Dropout (attn/relu/embed/final) | 0.0 / 0.2 / 0.1 / 0.1 | 0.2 / 0.3 / 0.2 / 0.2 |
| L2 reg ($t/a/v$) | 5e-4 / 1e-4 / 5e-4 | 5e-3 / 1e-4 / 5e-3 |
| Learning rate (main) | 2e-5 | 1.5e-6 |
| Learning rate (concept) | 1e-4 | 3e-3 |
| Learning rate ($t/a/v$ weights) | 8e-4 | 1e-4 |
| Gradient clip | 1.0 | 1.0 |

```
You are an expert in human intent recognition.
Your task is to analyze the following {num_concepts} concepts and
    decide whether they are relevant to the category {Complain} by a
    factor k between -1 and 1.
Concepts can be positively or negatively related to the category, so
    please consider both aspects.
Any Concepts that are useful for understanding the category
    {Complain} when coappear with other concepts should be considered
    relevant.
Here are the concepts:
{concept_text}
You should give a brief explanation of the factors in a
    human-readable format for each concept, strict in the format:
    1. [Original Concept 1]: k_1. Explanation for factor 1
    2. [Original Concept 2]: k_2. Explanation for factor 2
    ...

Please ensure your output is valid and well-formed, and the
    explanations are concise and clear.Make sure to include
    explanations for each factor.
```

## C DETAILED EXPERIMENTAL SETTINGS

We conduct experiments on two multimodal intent recognition benchmarks MIntRec and MIntRec2.0 with their official train/dev/test splits. Feature extraction relies on pretrained ResNet for video, Wav2Vec for audio, and BERT tokenizer for text, and both features and concept annotations will be released with the codebase. The task is framed as multi-class classification, and models are trained with gradient clipping, attention masking for unimodal encoders, and early stopping based on the accuracy on the validation set. To avoid premature convergence, a warm-up phase is employed before standard early stopping is activated. Additional hyperparameters are shown in Table 4 with batch size 16 for both datasets, learning rate 2e-5 for MIntRec and 1.5e-6 for MIntRec2.0, and concept set size 100 and 150 respectively. All methods are trained with seeds fixed from 0 to 4.

## D ADDITIONAL ABLATIONS ON CONCEPT SELECTION

To evaluate the effect of different strategies on concept selection, we conduct ablation studies on similarity filtering, submodular selection, and the number of concepts, with results shown in Table 5. Notably, in the w/o submodular selection setting, concepts are randomly selected to meet the required quantity. For varying concept numbers, we adjust the original 5 concepts per intent to obtain settings with either 8 or 3 concepts per intent. From the experimental results, we observe that both

Table 5: Ablation studies for concept selection on the MIntRec and MIntRec2.0 datasets.

| Ablations | MIntRec | | | | | | MIntRec2.0 | | | | | |
|---|---|---|---|---|---|---|---|---|---|---|---|---|
| | ACC (↑) | F1 (↑) | P (↑) | R (↑) | WF1 (↑) | WP (↑) | ACC (↑) | F1 (↑) | P (↑) | R (↑) | WF1 (↑) | WP (↑) |
| w/o similarity filtering | 74.97 | 72.15 | 73.26 | 72.25 | 74.92 | 75.72 | 60.37 | 54.41 | 56.54 | 54.18 | 59.47 | 59.81 |
| w/o submodular selection | 74.79 | 71.45 | 72.96 | 71.25 | 74.56 | 75.17 | 60.34 | 54.88 | 57.96 | 54.70 | 59.47 | 60.01 |
| 3 concepts per intent | 74.65 | 71.30 | 72.38 | 71.60 | 74.54 | 75.32 | 60.41 | 54.90 | 57.61 | 54.80 | 59.50 | 59.89 |
| 8 concepts per intent | 75.06 | 71.98 | 72.51 | 72.52 | 74.76 | 75.21 | 60.65 | 54.85 | 57.78 | 54.88 | 59.65 | 60.29 |
| Full (5 concepts per intent) | **75.91** | **73.06** | **73.48** | **73.37** | **75.76** | **76.20** | **60.89** | **55.46** | **58.05** | **55.22** | **60.04** | **60.46** |

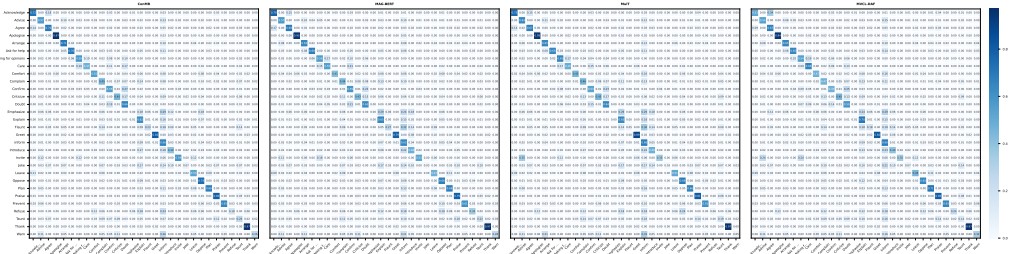

Figure 4: Confusion matrices of ConMR and three strong baselines on the MIntRec2.0 dataset.

removing similarity filtering and replacing the submodular selection strategy lead to a performance decline, causing a drop of over 0.5% on most metrics across both datasets. This indicates that our concept selection strategy effectively improves the quality of the concept set, preventing semantic redundancy among concepts and enhancing their discriminability. Moreover, while changing the number of concepts leads to performance decline, ConMR consistently surpasses most baselines, highlighting its robustness to concept quantity and its ability to maintain stable performance. Notably, reducing the number of concepts causes a larger drop than increasing them, likely because capturing complex intent semantics requires a sufficient set of concepts for effective representation.

## E    CLASSIFICATION PERFORMANCE ON MINTREC2.0

Similar to the analysis on MIntRec, we visualize confusion matrices of ConMR and the three strongest baselines on the MIntRec2.0 dataset to assess classification performance, as shown in Figure 4. Overall, ConMR achieves the highest accuracy in 15 out of 30 intent categories, clearly surpassing the baselines, which obtain the best results in only 7, 9, and 10 categories respectively. Consistent with the findings on MIntRec, this demonstrates the strong capability of ConMR to handle fine-grained and semantically overlapping categories. In particular, ConMR not only achieves excellent results on relatively straightforward intents such as *Thank*, *Praise*, and *Apologise*, but also delivers gains of more than 15% on more challenging categories such as *Taunt* and *Comfort*. This mirrors the conclusions drawn from MIntRec and further verifies the superiority of concept-level reasoning in recognizing complex intents. Apart from a drop in performance on classes like *Care*, ConMR exhibits consistently strong classification results overall. At the same time, it is worth noting that all four models, including ConMR, perform poorly on categories such as *Joke* and *Emphasize*. This suggests that in inherently ambiguous and highly challenging cases, ConMR has not yet shown a clear advantage, pointing to promising directions for future refinement.

## F    ADDITIONAL CASE STUDIES

To further illustrate the effectiveness and interpretability of ConMR, we present additional case studies from the MIntRec and MIntRec2.0 datasets, as shown in Figure 5 and Figure 6, where each case demonstrates how the model leverages multimodal concept activations for faithful intent prediction.

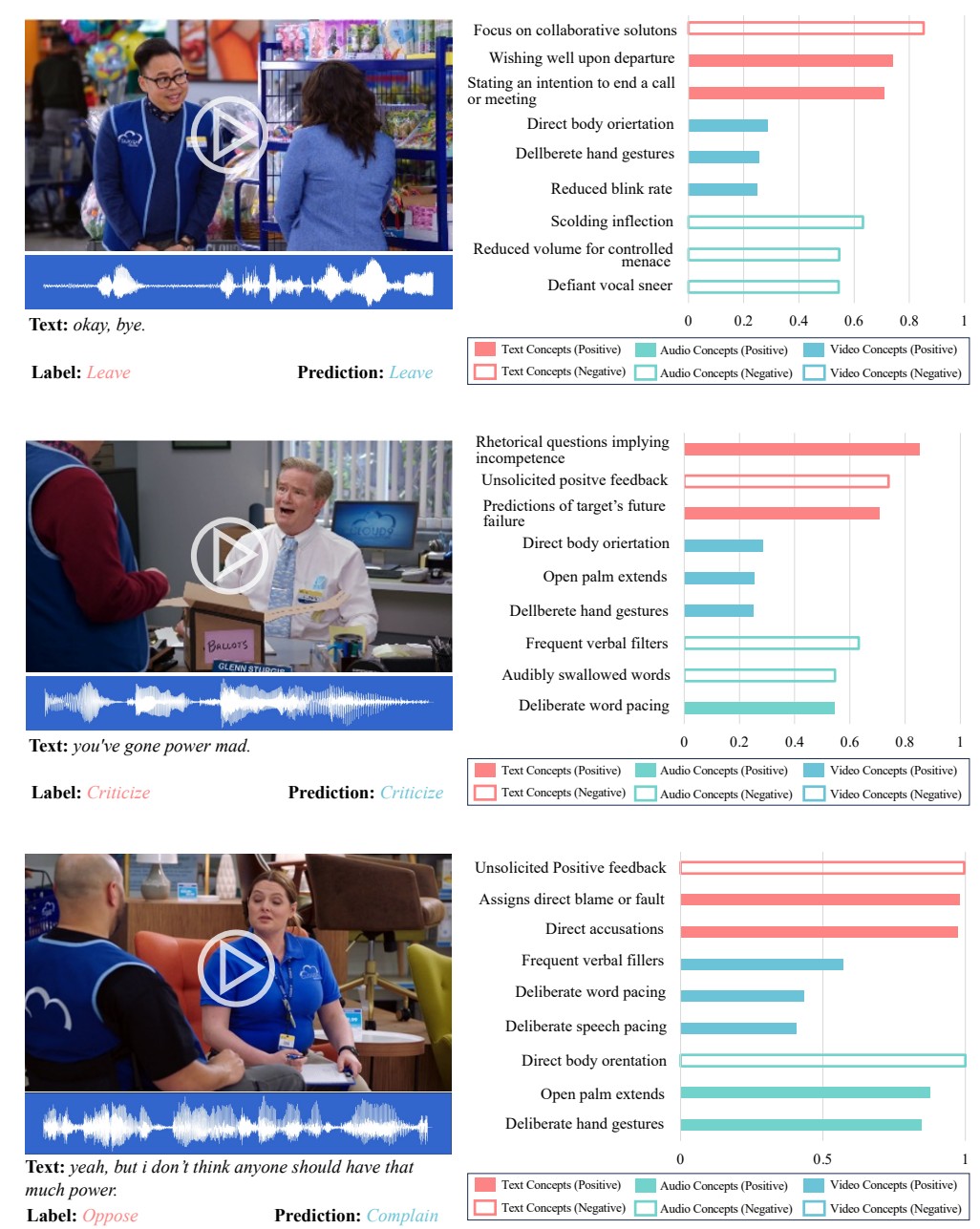

Figure 5: More cases on the MIntRec dataset.

## F.1 OVERALL ANALYSIS OF CONCEPT PERFORMANCE

In most cases, textual concepts provide the strongest signals, audio contributes complementary prosodic cues, and visual features serve as supportive context. This pattern underscores the central role of linguistic evidence, with prosody reinforcing emphasis and gestures grounding interpretation. Within each modality, a few concepts dominate because they frequently occur and are strongly linked to multiple intents. This dominance is less obvious in text, while audio and visual modalities often rely on repeatedly triggered cues. For instance, textual concepts like *Unsolicited positive feedback* often reach extreme activations close to $1$ or $-1$, since binary polarity is relatively easy to identify and highly influential. Audio cues such as *Frequent verbal fillers* are usually negatively activated, as scripted dialogue rarely contains natural disfluencies, while *Deliberate word pacing*

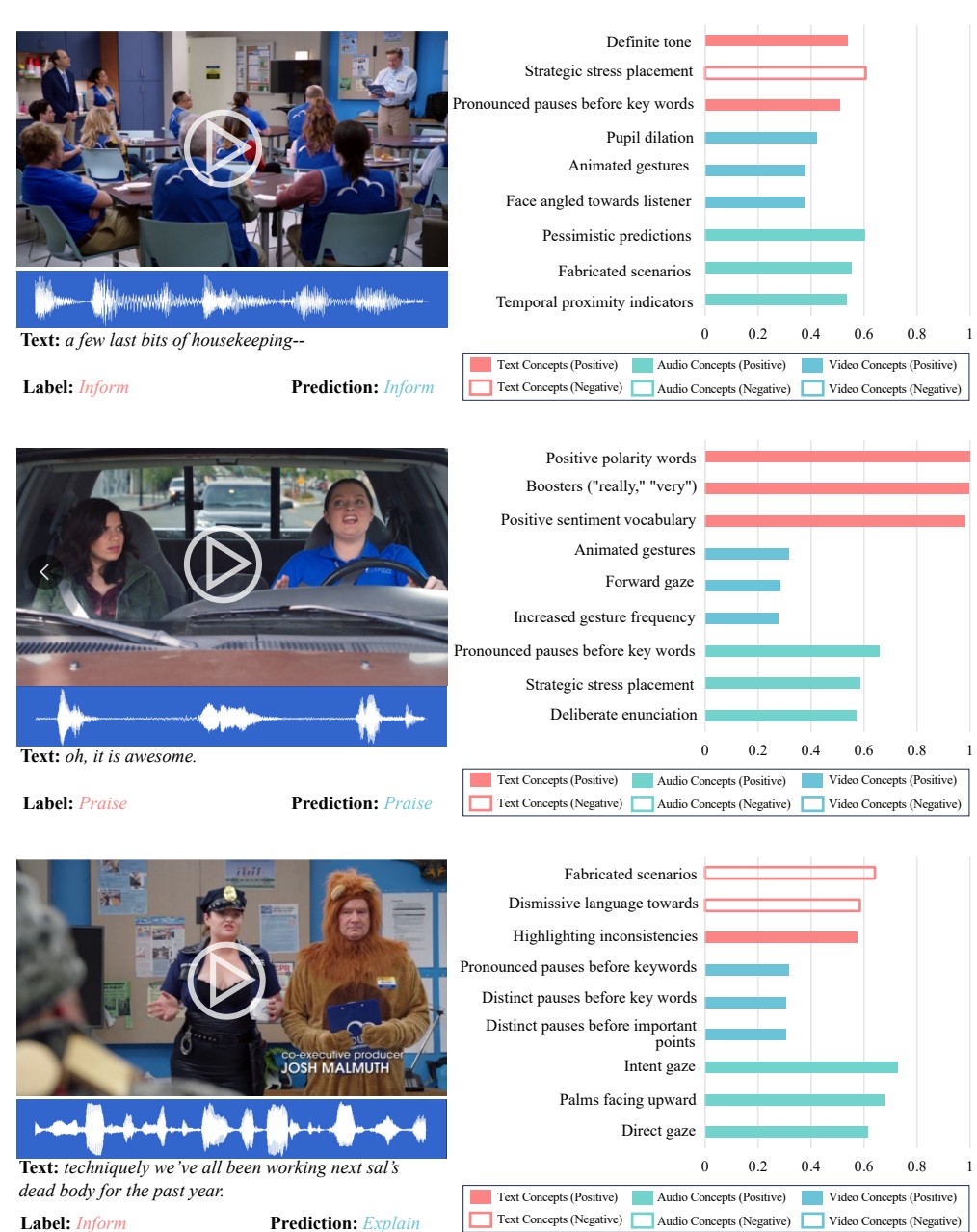

Figure 6: More cases on the MIntRec2.0 dataset.

is positively activated, reflecting prosodic emphasis typical of TV dialogue. For the MIntRec2.0 dataset, audio concepts such as *Pronounced pauses before key words* are strongly activated because they frequently enhance clarity. Visual concepts including *Open palm extends* or *Direct body orientation* repeatedly emerge as decisive, suggesting that a limited set of stable non-textual cues disproportionately shapes predictions. It is likely because non-textual annotations are less reliable than textual semantics, making the model rely more on a small group of discriminative cues.

## F.2 ANALYSIS OF SUCCESS PATTERNS

Across both MIntRec and MIntRec2.0, we observe recurring success patterns where ConMR effectively integrates multimodal concepts to capture intent. A first pattern is the accurate detection of

highly discriminative textual cues that directly convey the intended action or emotion. For example, in the first example of Figure 5, the cue *Wishing well upon departure* decisively supports the *Leave* intent, as "okay, bye" signals a clear closure. Besides, in the second example of Figure 5, *Rhetorical questions implying incompetence* and *Predictions of target's future failure* strongly anchor the *Criticize* intent with text of "you've gone power mad". Similarly, in Figure 6, concepts such as *Definite tone* with the original text of "a few last bits of housekeeping" highlight informative delivery, while *Positive polarity words and Boosters (e.g., "really", "very")* distinctly conveys praise along with the text of "oh, it is awesome".

Notably, the second pattern emerges when multiple modalities converge to reinforce the same interpretation. Prosodic cues such as *Deliberate word pacing* or *Pronounced pauses before key words* magnify sentiment or clarity, amplifying textual signals. For instance, deliberate pacing intensify the sharpness of critique in the *Criticize* case, while strategic pauses around "awesome" enhance positive affect in the *Praise* case. Visual concepts frequently act as stabilizers, with *Direct body orientation*, *Open palm extends*, and *Animated gestures* consistently reinforce stance and engagement. In the *Leave* case, mutual orientation and hand gestures captured non-verbal closure, while in the *Inform* case, animated explanations and attentive gazes embodied structured information delivery.

Taken together, these success cases reveal two dominant paradigms. First, intent is captured most reliably when the model identifies textual concepts with strong semantic alignment since textual information serves as the primary driver of intent recognition Zhang et al. (2022; 2024a). Second, complementary audio and visual cues enhance the robustness of predictions by reinforcing textual evidence and excluding confounding alternatives.

## F.3 ANALYSIS OF FAILURE PATTERNS

In contrast, failure cases expose situations where ConMR either misidentifies critical evidence or overweights concepts with limited discriminative power. One frequent failure pattern is the reliance on generic polarity concepts that cannot separate fine-grained negative intents. In the failure example of Figure 5, the textual cue *Unsolicited positive feedback* is strongly activated in negative contexts, yet it fails to distinguish between *Complain* and *Oppose*, both of which share negative polarity. In such cases, the model produces ambiguous predictions because it does not capture the more specific pragmatic distinctions. Similarly, prosodic features such as *Deliberate word pacing* or *Frequent verbal fillers* may dominate activations, but since scripted dialogue rarely contains natural disfluencies, these cues often add noise without clarifying intent.

Another failure pattern arises from spurious or erroneous activations that misrepresent the meaning of an utterance. In the failure example from MIntRec2.0 shown in Figure 6, the concept *Highlighting inconsistencies* is strongly activated for the line "technically we've all been working next to Sal's dead body for the past year." Although the model interprets this as explanatory, the utterance in fact conveys factual information, creating confusion between *Explain* and *Inform*. Negative concepts such as *Fabricated scenarios* and *Dismissive language towards others* are also triggered, further clouding the decision boundary. On the visual side, cues like *Open palm extends*, *Intent gaze*, and *Direct gaze* are activated but remain inconclusive, as such gestures are common across diverse intents.

Overall, these failure cases also highlight two major challenges. First, textual polarity cues may be overly simplistic to capture subtle distinctions between closely related intents. Second, non-textual concepts are more vulnerable to spurious activations that mislead the model. Addressing these issues requires refining concept definitions, improving annotation quality in audio and visual modalities, and designing mechanisms to control the weight of unreliable activations.

## G ANALYSIS OF CONCEPT ACTIVATION SCORE

To further analyze concept activations, we compute the KL divergence between the activation patterns predicted by ConMR and the ground-truth activations obtained from pretrained models on the test sets of MIntRec and MIntRec2.0. The results are shown in Figure 7 and Figure 8, where smaller absolute divergence values indicate more faithful alignment between learned and true concept activations, thereby reflecting the accuracy of concept-level modeling. Note that the x-axis in the figures represents different concepts, which are grouped by modality.

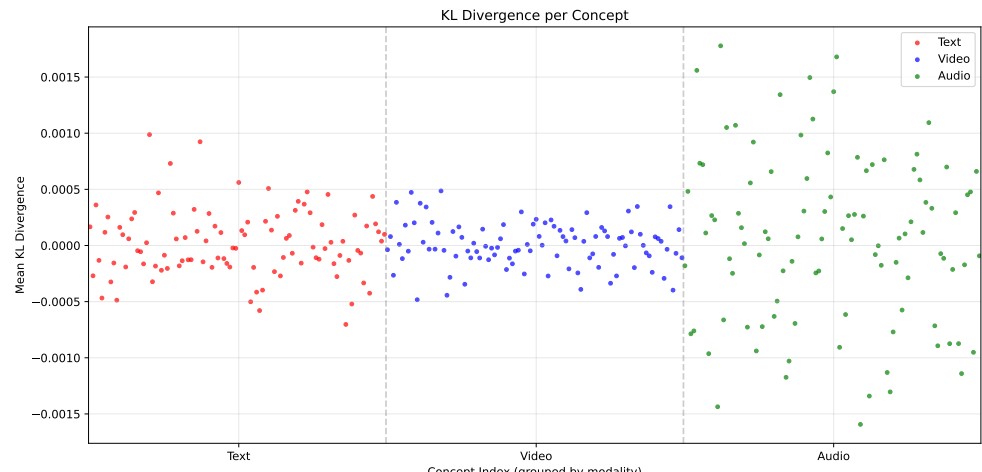

Figure 7: Concept activation score distribution on the MIntRec dataset.

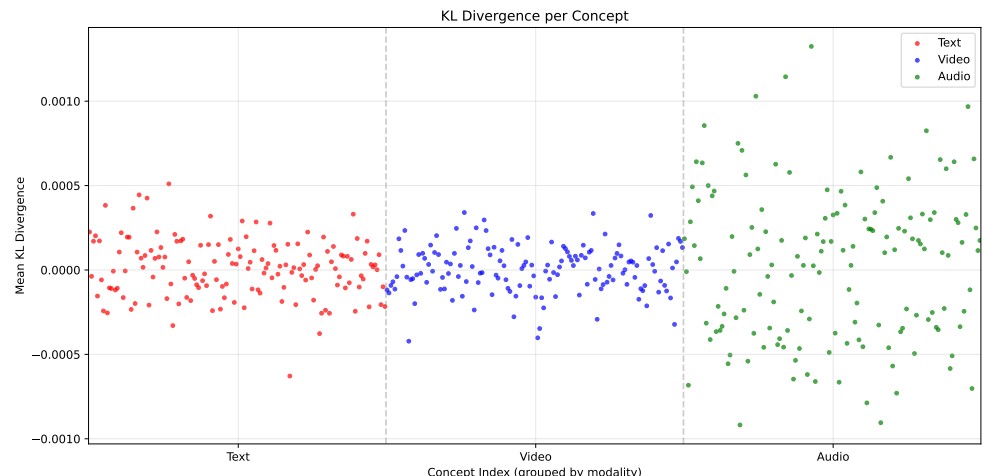

Figure 8: Concept activation score distribution on the MIntRec2.0 dataset.

From the results, we observe a broadly consistent alignment between the predicted and ground-truth concept activation patterns across both datasets. Text and video concepts exhibit consistently tighter alignment, with absolute KL divergence values typically below 0.0005, suggesting that ConMR not only recognizes but also precisely models the diverse manifestations of these concepts across samples. Although the alignment of audio concepts is comparatively weaker, their absolute KL divergence values remain within 0.001, still reflecting a reasonable degree of semantic consistency and demonstrating that ConMR is able to capture useful audio cues despite the higher variability in this modality. These results indicate that ConMR effectively models concepts across modalities while revealing inherent modality-specific differences, with text and video serving as stable semantic anchors and audio remaining more variable yet still informative, thereby providing a solid foundation for concept-level multimodal reasoning.

## H  LIMITATIONS

The first limitation of ConMR lies in its dependence on the quality of the concept sets. Although we adopt filtering strategies to ensure diversity and semantic validity, the framework may be weakened if the generated concepts are poorly aligned with target intents. As demonstrated in Appendix F, ConMR provides effective concepts which are decisive for reasoning, but adaptive refinement strate-

gies remain an important direction for future work. Second, not all intents can be adequately captured by simple concept activation patterns. While ConMR models most cases successfully in Appendix G, some intents may correspond to multiple complex activation modes that require further investigation. Finally, despite strong results on two challenging benchmarks, broader validation across diverse domains and application scenarios can be conducted to fully establish the generality and robustness of ConMR.

## I   THE USE OF LARGE LANGUAGE MODELS

In this work, we employ large language models (LLMs) solely as auxiliary tools. Specifically, LLMs are used in our method to generate candidate concepts and compute intent relevance scores, as detailed in Appendix B. In addition, we use LLMs during paper preparation exclusively for language polishing. LLMs don't contribute to research ideation or methodological design.

