# OpenReview forum: "Concept-level Multimodal Reasoning via Semantic Representation for Intent Recognition"
_ICLR.cc/2026/Conference — Submitted to ICLR 2026_

### Official Review · Reviewer_HBfx · 2025-10-31

**Soundness:** 3
**Presentation:** 2
**Contribution:** 3
**Rating:** 4
**Confidence:** 2

**Summary:**

This paper proposes a Concept-level Multimodal Reasoning framework for intent recognition named ConMR. Specifically, ConMR includes two core modules: Concept Representation Learning and Concept-level Multimodal Reasoning. The Concept Representation Learning module leverages a large language model to automatically generate and filter intent-related concepts for text, video, and audio modalities, and maps multimodal features into a unified, interpretable concept space under activation alignment supervision from pretrained modality-specific encoders. The Concept-level Multimodal Reasoning module models both concept-to-intent relevance and inter-concept relations derived from activation patterns, enabling structured reasoning paths from concepts to intents.

The experimental evaluation in this paper assessed the performance of the proposed ConMR on two challenging multimodal intent recognition benchmarks (MIntRec and MIntRec2.0), comparing it with various state-of-the-art methods. The results indicate that the proposed ConMR achieved consistent performance improvements and superior interpretability, with notable gains in accuracy and F1-score across diverse intent categories.

**Strengths:**

- The attempt to transform feature-level patterns into the explicit concept-level paradigm is valuable.
- The experimental analysis is overall thorough; even the appendix is well-organized.

**Weaknesses:**

- The quality of the entire concept space and the evaluation of intent relevance rely heavily on the capability and stability of the large language model. If the LLM’s understanding is inaccurate in specific domains or performs poorly in cross-lingual and cross-cultural scenarios, it may introduce noisy concepts and biased relevance scores.
- The ablation study on the concept selection strategy is somewhat incomplete. For example, whether similarity filtering is necessary, whether submodular selection is applied, or how varying the number of concepts affects performance.
- The paper lacks discussion on why SBERT, XCLIP, and CLAP were specifically chosen to compute association scores.
- Concept-level reasoning mainly focuses on constructing and reasoning within the same modality; however, cross-modal concept connections often better reflect real intent (e.g., contradictions or alignments between visual expression and verbal tone). The current fusion mainly relies on concatenation and shared weights, lacking explicit modeling of cross-modal concept interactions.
- Considering that the model itself requires an LLM and involves multiple encoders (SBERT, XCLIP, CLAP, PLM, Swin Transformer, WavLM), there is a complex coupling between these components, making it difficult to analyze the contribution of each model or conduct ablations on different combinations. Moreover, the combination of multiple models introduces additional computational burdens.

**Questions:**

- It seems that the concept set is usually fixed before training and cannot be dynamically adjusted based on samples or contexts. Does this imply that a separate model must be prepared for each benchmark, thereby posing challenges for open-domain scenarios?

---

> ### Author Response · Authors · 2025-11-21
> **Response to Reviewer HBfx (Part 1/2)**
>
> **Reply to Weaknesses:**
>
> **A1** Thank you for raising the concern. To ensure the reliability of concept generation and intent relevance scoring, we follow [1],[2] and design a complete Concept Pipeline (detailed in Appendix B). This pipeline includes task-specific prompt design, removal of concepts that are too close to intents, removal of redundant concepts, and submodular selection based on discriminability and coverage score. These steps ensure that the final concepts are highly relevant to the target intent, and semantically non-redundant. On this basis, we further ask the LLM to assign relevance scores along with corresponding explanations, which enhances the reliability of the scores. Moreover, our multimodal reasoning mechanism employs a dual-path strategy combining concept-to-intent and inter-concept reasoning together with a weighting mechanism, which improves robustness against noisy concepts and potentially biased relevance scores in challenging scenarios.
>
> **A2** Thank you for your comments regarding the ablations on the concept selection strategy. In Section 5.1, we performed ablation studies on ConMR’s key modules to demonstrate the contribution of each component. To address your concern, we further conducted additional ablations on the concept generation process, including similarity filtering, submodular selection, and concept quantity on both datasets. For the varying concept quantity setting, we ablate ConMR using 3 and 8 concepts per intent while the original setting uses 5, and report the corresponding total concept counts for each dataset in the table.
>
> The results show that removing similarity filtering causes performance drops exceeding 0.5% across most metrics on both datasets, highlighting its role in eliminating concepts too close to the intent and producing a higher-quality concept set. Besides, ablating submodular selection also results in a clear performance drop, demonstrating that balancing discriminability and coverage is essential for maintaining a diverse and informative concept set. By contrast, increasing the number of concepts has  minimal effect, with performance drops smaller than those observed in the previous two ablations across most metrics. Moreover, although decreasing the number of concepts leads to some degradation on MIntRec, the overall performance remains strong relative to most baselines. These results indicate that ConMR is robust to concept quantity while maintaining stable and strong performance.
>
> |MIntRec|Acc|F1|P|R|WF1|WP|
> |-|:-:|:-:|:-:|:-:|:-:|:-:|
> |w/o similarity filtering| 74.97 | 72.15 | 73.26 | 72.25 | 74.92 | 75.72 |
> |w/o submodular selection| 74.79 | 71.45 | 72.96 | 71.25 | 74.56 | 75.17 |
> |ConMR (3 concepts per intent)| 74.65 | 71.30 | 72.38 | 71.60 | 74.54 | 75.32 |
> |ConMR (8 concepts per intent)| 75.06 | 71.98 | 72.51 | 72.52 | 74.76 | 75.21 |
> |Full (5 concepts per intent)| **75.91**  | **73.06** | **73.48** | **73.37** | **75.76**  | **76.20** |
>
> |MIntRec2.0|Acc|F1|P|R|WF1|WP|
> |-|:-:|:-:|:-:|:-:|:-:|:-:|
> |w/o similarity filtering| 60.37| 54.41| 56.54| 54.18| 59.47| 59.81|
> |w/o submodular selection| 60.34 | 54.88 | 57.96 | 54.70 | 59.47 | 60.01 |
> |ConMR (3 concepts per intent)| 60.41| 54.90| 57.61 | 54.80 | 59.50 | 59.89  |
> |ConMR (8 concepts per intent)| 60.65 | 54.85 | 57.78 | 54.88 | 59.65 | 60.29 |
> |Full (5 concepts per intent)| **60.89**  | **55.46** | **58.05** | **55.22** | **60.04**  | **60.46** |
>
> **A3** We selecte SBERT, XCLIP, and CLAP because they represent the mainstream and most widely adopted cross-modal alignment encoders in their respective domains, demonstrating strong empirical performance and extensive use in both CBMs [3] and multimodal alignment field [4] [5]. Specifically, SBERT serves as the standard for sentence-level semantic similarity, with various text-based CBMs leveraging embeddings to evaluate concept activation. XCLIP is a leading video–text model, widely applied in multimodal alignment tasks and recent CBM variants for video understanding. CLAP is a prominent audio–text similarity encoder, supporting numerous audio grounding, captioning, and acoustic concept extraction. Their broad adoption and robust cross-modal alignment capabilities make them well-suited for grounding concept activations in ConMR, enabling precise mapping of textual concept semantics to features in text, video, and audio. This ensures activation scores accurately reflect concept presence, supporting interpretable and semantically faithful concept-level reasoning across modalities.

---

> > ### Author Response · Authors · 2025-11-21
> > **Response to Reviewer HBfx (Part 2/2)**
> >
> > **A4** Thank you for your insightful review. We clarify that the dual concept-level multimodal reasoning paths in ConMR are designed to unify reasoning over fine-grained semantic concepts extracted from different modalities. Notably, these concepts serve as more fundamental and finer-grained semantic units than modality-level features, reflecting the key cues from which multimodal intent inherently arises and thus transcending modality-specific differences. By reasoning directly over these concepts through both concept-to-intent and inter-concept pathways, ConMR captures the underlying semantic structure at a deeper level, enabling more coherent and semantically faithful intent understanding than approaches that rely solely on explicit cross-modal interaction modeling. This also highlights the key advantage of concept-level reasoning, enabling more flexible and expressive understanding of multimodal complex semantics.
> >
> > **A5** Thank you for raising the concern. We clarify the roles, ablation considerations, and computational cost of the LLM and multiple encoders in ConMR as follows:
> >
> > (1) Roles of the LLM and encoders. Aligned with [1], ConMR incorporates an LLM and two types of encoders. The leading LLM (Gemini-2.5 Flash) is employed for high-quality concept generation. One set of encoders (SBERT, XCLIP, CLAP) is used to align concept activations and represents mainstream cross-modal alignment models widely adopted in CBM research [1]. The other set (PLM, Swin Transformer, WavLM) provides multimodal feature embeddings and corresponds to modality-specific encoders commonly used in multimodal intent understanding [6] [7]. All of these encoders are essential for constructing robust and semantically meaningful concept representations.
> >
> > (2) Ablation considerations. These encoders represent state-of-the-art solutions in their respective domains, demonstrating strong generalization and validated contributions in CBMs and multimodal intent tasks. Since ConMR focuses on concept-level multimodal reasoning, these models serve primarily as foundational support. For this reason, we do not conduct ablations on them in the paper. Furthermore, because the processing of each model is largely independent, individual components can be replaced or removed to perform targeted ablations without disrupting the overall method.
> >
> > (3) Computational cost. Concept generation, activation alignment, and modality-specific feature encoding are performed as preprocessing steps and do not require model training. Consequently, despite incorporating multiple pretrained models, the computational cost of ConMR remains comparable to that of other multimodal baselines and CBMs.
> >
> > **Reply to Questions:**
> >
> > **A6** Thank you for your insightful comment. While the concept set in ConMR is initially generated and fixed, it can be dynamically adapted to the task context and is not tied to any specific benchmark. As detailed in Appendix B, concept generation relies only on the task background and labels, without requiring access to individual samples, enabling the concept set to generalize across different benchmarks within the same task domain. Moreover, for open-domain scenarios, the concept set can be efficiently expanded or updated, and new concepts can be incorporated with lightweight fine-tuning to address novel tasks effectively.
> >
> > **References**
> >
> > [1] Label-Free Concept Bottleneck Models. Tuomas Oikarinen et al. ICLR 2023.
> >
> > [2] Language in a Bottle: Language Model Guided Concept Bottlenecks for Interpretable Image Classification. Yue Yang et al. CVPR 2023.
> >
> > [3] Concept Bottleneck Large Language Models. Chung-En Sun et al. ICLR 2025.
> >
> > [4] Bidirectional Cross-Modal Knowledge Exploration for Video Recognition with Pre-trained Vision-Language Models. Wenhao Wu. CVPR 2023.
> >
> > [5] Large-Scale Contrastive Language-Audio Pretraining with Feature Fusion and Keyword-to-Caption Augmentation. Yusong Wu. ICASSP 2023.
> >
> > [6] Token-Level Contrastive Learning with Modality-Aware Prompting for Multimodal Intent Recognition. Qianrui Zhou et al. AAAI 2024.
> >
> > [7] Multimodal Classification and Out-of-distribution Detection for Multimodal Intent Understanding. Hanlei Zhang et al. IEEE Transactions on Multimedia.

---

> > > ### Comment · Reviewer_HBfx · 2025-11-28
> > >
> > > I have read the authors’ response. I appreciate that the extra ablation experiments for the key components, and I accept the authors' explanation regarding LLM bias.
> > >
> > > However, the discussion on the choice of multiple encoders and the way they are combined is still insufficient. In addition, although these additional models do not require extra training, they are still involved in the inference pipeline and would affect the model’s FLOPs and actual latency.
> > >
> > > Moreover, the ConMR is based on a fixed concept set, which would require additional adjustments in open-domain scenarios, which somewhat limits the generalizability of the approach.
> > >
> > > Therefore, I have decided to keep my score as 4.

---

### Official Review · Reviewer_dY2Z · 2025-10-31

**Soundness:** 3
**Presentation:** 3
**Contribution:** 3
**Rating:** 6
**Confidence:** 3

**Summary:**

This paper presents ConMR, a novel framework for multimodal intent recognition that elevates reasoning from the feature level to the concept level. The core idea is to leverage LLM-generated concepts as semantic anchors, learn explicit concept representations from multimodal features, and then perform structured reasoning over concept-intent and inter-concept relations. The authors demonstrate state-of-the-art performance on two established benchmarks (MIntRec and MIntRec2.0) and provide extensive experiments, including ablation studies and case analyses, to validate their design choices.

**Strengths:**

This paper's principal strengths lie in its conceptual contribution. It proposes a concept level reasoning approach for multimodal intent recognition, effectively bridging the gap between low-level features and high-level intents to enhance both performance and, crucially, model interpretability. The proposed ConMR framework is meticulously designed, integrating LLM-based automatic concept generation, a supervised feature-to-concept transformation, and a dual-path reasoning module that models both concept-to-intent and inter-concept relations. This rigorous design is supported by compelling empirical evidence, including state-of-the-art results on two benchmarks, ablation studies that validate each component, and case analyses that demonstrate its transparent decision-making. Furthermore, the framework's robustness is confirmed by its consistent performance across different LLM, solidifying its value as a significant advancement towards trustworthy multimodal AI.

**Weaknesses:**

see questions

**Questions:**

1. How would the framework perform in a low-resource setting where access to powerful LLMs like Gemini-2.5 is limited? Is there a fallback strategy or a lighter-weight alternative for concept generation?
2. The failure case analysis in Appendix E.3 is excellent. Could the framework be extended to incorporate a feedback loop that uses such mis-predictions to iteratively refine the concept set?

---

> ### Author Response · Authors · 2025-11-21
> **Response to Reviewer dY2Z**
>
> **A1** Thank you for raising the concern. To evaluate ConMR’s performance in low-resource settings, we replace the powerful Gemini-2.5 Flash with the leading LLM with lighter weight (Qwen3-8B) and conduct experiments on the MIntRec dataset. The results demonstrate that ConMR with Qwen3-8B delivers performance nearly on par with the Gemini-2.5 Flash setting, with only minimal differences across most metrics and even outperforming it on P metric. This strong performance under a substantially lighter LLM highlights ConMR’s intrinsic robustness, driven by its explicit concept representations and dual-path multimodal reasoning, which reduce the influence of lower-quality concepts, preserve semantic fidelity, and sustain reliable intent inference.
>
>
> |MIntRec|Acc|F1|P|R|WF1|WP|
> |-|:-:|:-:|:-:|:-:|:-:|:-:|
> |ConMR with Qwen3-8B| 75.15  | 72.79 | **74.41** | 72.56 | 75.03  | 75.90 |
> |ConMR with Gemini-2.5 Flash| **75.91**  | **73.06** | 73.48 | **73.37** | **75.76**  | **76.20** |
>
> **A2** Thank you for your appreciation of our work. We agree that incorporating a feedback loop to iteratively refine the concept set based on mispredictions is a promising direction for future improvement. ConMR’s core strength lies in constructing explicit concept representations and performing concept-level multimodal reasoning, and we propose the following feasible design for such an extension.
>
> After each training epoch, we can apply a concept refinement mechanism to iteratively optimize the concept set based on misprediction samples. For each concept, we compute the concept hallucination rate, defined as the fraction of cases where the concept is highly activated among all error cases, which indicates potential spurious or misleading cues. Specifically, it is calculated as the number of error cases where the concept is highly relevant divided by the total number of cases where the concept is highly relevant. Similarly, the concept utility quantifies the concept’s contribution to correct predictions and is computed as the number of correct predictions where the concept is highly relevant divided by the total number of predictions in the epoch, identifying redundant or low-impact concepts. Concepts with high hallucination rates and low utility are prioritized for removal. These are then replaced by generating new high-quality concepts through the concept generation pipeline described in Appendix B. The model then continues training in the next epoch based on the revised concept set, allowing the set to dynamically evolve and continuously improve concept effectiveness, thereby enabling more accurate and robust concept-level multimodal reasoning.

---

### Official Review · Reviewer_XnHa · 2025-11-03

**Soundness:** 1
**Presentation:** 1
**Contribution:** 2
**Rating:** 2
**Confidence:** 3

**Summary:**

The paper introduces ConMR, a concept-level multimodal reasoning model that uses LLM-generated concepts and relevance scores to supervise intent recognition training. The framework is clearly presented and shows consistent empirical improvements, with potential benefits in reducing inference time and cost compared to full MLLMs.

However, its true novelty is minimal—the method mainly distills MLLM priors through an additional MSE loss, while the underlying structure remains a standard CBM.
The claimed "reasoning" ability is in fact semantic alignment, not autonomous reasoning, and the paper overstates its originality.
Therefore, this is a practical but not theoretically innovative work whose main value may lie in efficiency rather than conceptual contribution.

**Strengths:**

1. The paper proposes a two-stage intent understanding framework consisting of Concept Representation Learning and Concept-level Multimodal Reasoning. It transforms entangled multimodal features into explicit concepts and reasons over their relations, using LLM-generated semantic concepts as interpretable anchors. This design improves interpretability, enabling the model to produce concept-level activations that correspond to human-understandable semantic units.
2. The experiments are conducted on two widely used benchmarks (MIntRec and MIntRec2.0), and the results are consistent across multiple evaluation metrics. The proposed method achieve the best performance compared with fusion-based methods and contrastive learning based methods.
3. The manuscript is logically structured, clearly written, and easy to follow despite some overcomplicated exposition.

**Weaknesses:**

1. Given that the proposed method already relies on LLMs for concept supervision and intent-relevance supervision, it is unclear why the authors do not directly employ LLMs for multimodal intent recognition (e.g., via zero-shot or fine-tuning approaches) instead of introducing additional concept reasoning layers.
Importantly, the experiments do not include comparisons with LGSRR[1], which, according to the authors, is the first work leveraging LLMs for multimodal intent recognition in Line 54.
This paper only compares with traditional fusion-based methods (MulT, MAG-BERT, MISA, etc.) and contrastive approaches (TCL-MAP, MVCL-DAF, etc.), but lacks LLM-based baselines, including:
* MLLM zero-shot/few-shot baselines, e.g., Gemini-2.5 directly predicting intents from multimodal inputs;

* MLLM pseudo-labeled baselines, where MLLMs label training samples with / without CoTs and a lightweight model is then trained on those labels.

Such comparisons should be included to convincingly demonstrate that the improvement of ConMR comes from its structured semantic reasoning mechanism rather than merely relying on LLM priors.

[1] Llm-guided semantic relational reasoning for multimodal intent recognition, Qianrui Zhou, et. al. 2025. arXiv

2. Modern multimodal LLMs (e.g., Gemini 2.5, Qwen-VL-Series) are already capable of: (1) directly handling text, audio, and visual inputs,(2) producing explicit reasoning chains via prompting, and (3) generating concept-level explanations. Therefore, ConMR’s approach appears to distillate LLM capabilities to ConMR’s training stage.
The authors need to clarify what concrete advantages the ConMR provides in terms of performance or interpretability beyond what a capable MLLM can already achieve.

3. This is a concern of the motiation.

Lines 56–61:

> "First, existing methods predominantly operate at the feature level, relying on entangled and abstract representations that leave a substantial gap between low-level multimodal signals and the nuanced semantics of human intent. Second, they lack explicit and structured multimodal reasoning mechanisms capable of modeling the interplay between high-level semantic representations, which makes it difficult to construct transparent and discriminative paths that bridge raw inputs to complex intents. "

However, these challenges are no longer valid in the LLM era, since current LLMs already perform feature disentanglement and explicit chain-of-thought reasoning for multimodal tasks. Thus, this argument cannot serve as a justification for proposing ConMR.

Additionally, lines 54–56:

> "while LGSRR Zhou et al. (2025) represents the first attempt to leverage Large Language Models (LLMs) to guide multimodal intent recognition. "

If so, the authors must clearly articulate what limitations of LGSRR or other LLM-based approaches remain unsolved.
The two points raised in lines 56–61 describe issues of pre-LLM fusion methods, not of LLM-based methods.
Therefore, the introduction currently fails to identify the real gap that ConMR aims to address in the post-LLM context.

4. The novelty is overstated.

Line 136-138:

> " Although CBM research has made notable strides in interpretability, it remains confined to surface-level contribution scores of concepts rather than capturing their intrinsic semantics, which fundamentally constrains its performance."

This claim is somewhat overstated.The real methodological novelty of ConMR lies only in introducing LLM-generated concept–intent relevance scores as explicit supervision via the MSE loss. Traditional Concept Bottleneck Models (CBMs) indeed learn a concept layer followed by a linear classifier to map the concepts and intent labels, where concepts are usually human-annotated or similarity-based matching. ConMR differs mainly in that: (1) the concept supervision is generated by an LLM (2) the concept-to-intent supervision also comes from the LLM’s semantic similarity scores. Therefore, for the method proposed in this paper, the structure is CBM-style, while the semantics come from the LLM. It is also explicitly shown:

* KL $\rightarrow$ concept-semantic consistency
* MSE $\rightarrow$ LLM-prior consistency
* CE $\rightarrow$ task-label consistency

Only the LLM-based semantic supervision is genuinely new. This is a practical innovation, not a theoretical one. It can be describe as a LLM-guided CBM method. This paper over-claims its methodological novelty.

5. Ablation results largely reflect dependence on LLM priors rather than genuine reasoning ability. In Table 2, the ablation outcomes can be directly explained by removing or preserving access to the LLM-derived priors.

Line 369-372:

> "Besides, a severe degradation is observed when the learnable transformation W is replaced with linear layers with activations (w/o W), with metrics on MIntRec2.0 dropping by more than 7%, which highlights the critical role in generating robust concept representations."

Line 372-374:

> "In the concept-level multimodal reasoning module, removing LMSE (w/o LMSE) results in performance drops from 0.49% to 2.28% across all metrics on both datasets, confirming the importance of LLM-based intent relevance score supervision."

Line 374-375:

> "Furthermore, ablating the concept-to-intent pathway (w/o Zconcept) causes a severe collapse, with accuracy on MIntRec dropping to 36.36% and F1 on MIntRec2.0 falling to 46.99%."

W learns LLM-provided "semantic ground truth", and it is essentially a distillation of the LLM prior, not a discovery of new relationships among the data.

Similarly, line 372-375:

> "For concept-to-intent reasoning, intent-conditioned relevance scores generated by Gemini-2.5 are leveraged to guide a weighting network through MSE loss to selectively reinforce concept features."

It shows that removing $𝓛_{MSE}$ or the concept-to-intent pathway $Z_{concept}$ leads to drastic performance drops (up to total collapse). The fact may be that removing these components effectively removes the LLM semantic channel. Therefore, the ablation study does not demonstrate autonomous concept-level reasoning, however, it shows that ConMR’s performance is driven by dependence on LLM-generated priors rather than self-learned reasoning capacity.

**Questions:**

NA

---

> ### Author Response · Authors · 2025-11-21
> **Response to Reviewer XnHa (Part 1/3)**
>
> **A1** We will address your concerns from the following aspects.
>
> (1) Limitations of directly employing LLMs. Despite recent progress, current LLMs still face fundamental challenges in multimodal intent recognition. Existing models excel at perceiving and grounding multimodal inputs (e.g., extracting objects and characters) and can perform simple reasoning steps based on commonsense knowledge (e.g., concept generation and relevance analysis), but they lack the capability to reliably leverage these multimodal cues to infer high-level cognitive intents [1], which arises from the inherently complex nature of intent as a high-level semantic construct. Besides, while supervised fine-tuning (SFT) can significantly improve performance [1], it demands substantial computational resources and extended training time, limiting the practicality for real-world deployment.
>
> (2) Strengths of concept reasoning layers. The core advantages of our method lie in shifting multimodal reasoning from opaque feature-level patterns to an explicit concept-level paradigm, comprised of Concept Representation Learning and Concept-level Multimodal Reasoning. In this framework, LLMs are used only to generate candidate concepts that indicate potential multimodal cues and provide auxiliary relevance signals, while the concept reasoning layers perform the core reasoning by transforming multimodal inputs into semantically interpretable concept representations and establishing a structured reasoning pathway from concepts to intents, addressing the aforementioned limitation of LLMs.
>
> (3) Comparison with LLM-based baselines. To address the concern, we include comparisons with MLLM baselines (Gemini-2.5 Flash under zero-shot, few-shot and CoT settings) and an MLLM pseudo-labeled baseline (LGSRR [2]) on both datasets. For few-shot setting, we provide one representative example per intent class as in-context demonstrations, while for the CoT setting, we guide the model through a structured reasoning path that includes global perception, key-concept identification, relational analysis, and intent inference. As shown in the results, ConMR consistently outperforms all LLM-based baselines, demonstrating that its gains arise from structured semantic reasoning rather than LLM priors. Gemini-2.5 Flash performs poorly under both zero- and few-shot settings, with gaps exceeding 10% across all metrics, confirming that even state-of-the-art MLLMs struggle to infer high-level intents directly. Besides, as the first LLM-based method for multimodal intent recognition and a recent baseline, LGSRR still trails ConMR by over 1% on MIntRec and remains slightly behind on MIntRec2.0, further highlighting the advantage of our concept-level reasoning paradigm.
>
> |MIntRec|Acc|F1|P|R|WF1|WP|
> |-|:-:|:-:|:-:|:-:|:-:|:-:|
> |Gemini-2.5 Flash Zero-shot| 55.12  | 52.46 | 56.03 | 54.19 | 55.74  | 56.30 |
> |Gemini-2.5 Flash Few-shot| 61.44  | 59.91 | 60.04 | 56.77 | 60.18  | 62.52 |
> |Gemini-2.5 Flash CoT| 66.52  | 63.03 | 65.52 | 64.88 | 67.74  | 65.45 |
> |LGSRR| 73.26  | 70.77 | 72.08 | 70.07 | 72.97  | 73.15 |
> |ConMR| **75.91**  | **73.06** | **73.48** | **73.37** | **75.76**  | **76.20** |
>
> |MIntRec2.0|Acc|F1|P|R|WF1|WP|
> |-|:-:|:-:|:-:|:-:|:-:|:-:|
> |Gemini-2.5 Flash Zero-shot| 40.93  | 41.35 | 42.67 | 39.82 | 42.48  | 46.09 |
> |Gemini-2.5 Flash Few-shot| 43.15  | 42.27 | 42.06 | 40.53 | 45.78  | 47.81 |
> |Gemini-2.5 Flash CoT| 44.19  | 42.25 | 44.58 | 44.39 | 44.33  | 48.78 |
> |LGSRR| 60.30  | 55.04 | 57.57 | 54.17 | 59.77  | 60.20 |
> |ConMR| **60.89**  | **55.46** | **58.05** | **55.22** | **60.04**  | **60.46** |
>
>
> **A2** Our method outperforms MLLMs in two key aspects:
>
> (1) Performance. ConMR significantly outperforms modern MLLMs under zero-shot and few-shot settings for multimodal intent recognition by employing explicit concept-level multimodal reasoning that capture the diverse manifestation patterns of high-level cognitive intents. While MLLMs can process multimodal inputs and exhibit strong reasoning ability, ConMR delivers more reliable and accurate intent recognition, as evidenced by comparisons with LLM-based baselines.
>
> (2) Interpretability. Although MLLMs can generate reasoning chains and concept-level explanations, their reliability in inferring high-level semantic intents is limited [1], and the inconsistencies between reasoning paths and final predictions may compromise interpretability [3]. In contrast, ConMR employs explicit semantic concepts as the fundamental reasoning units and captures intents through a dual inference process linking concepts to intents and modeling inter-concept interactions. At each stage, the reasoning is supported by semantically meaningful concept representations and clear relational evidence, such as relevance and activation scores, ensuring that the process is both transparent and reliable. Furthermore, the case study in Section 5.3 provides an intuitive demonstration of how ConMR performs concept-level reasoning.

---

> > ### Author Response · Authors · 2025-11-21
> > **Response to Reviewer XnHa (Part 2/3)**
> >
> > **A3** Despite the significant progress of LLMs, they still face substantial challenges in achieving reliable feature disentanglement and explicit chain-of-thought reasoning for multimodal tasks involving complex semantics. For feature disentanglement, prior studies [4],[5] have shown that existing LLMs still fail to achieve reliable disentanglement of multimodal features, often hallucinating non-existent objects or misidentifying key concepts in the input, indicating that their internal representations remain entangled and semantically unstable. Regarding explicit chain-of-thought reasoning, in the **A1** CoT setting where Gemini-2.5 Flash is guided through global perception, key-concept identification, relational analysis, and intent inference, it still falls short of our method and other baselines, which underscores that leading LLMs struggle to construct transparent reasoning paths for complex multimodal intents.
> >
> > Besides, to the best of our knowledge, LGSRR constitutes the first and only framework that incorporates LLM guidance into multimodal intent recognition, yet it still exhibits notable limitations in these aspects. Specifically, it relies on three coarse-grained and inherently entangled intermediate semantic concepts derived from LLM-generated descriptions, which suffer from information loss and hallucinations. Furthermore, its reasoning mechanism inadequately captures the mapping from key semantics to intent, constrained to predefined logical relations and lacking sufficient coverage for complex multimodal scenarios.
> >
> >
> > **A4** The main novelty of ConMR is twofold:
> >
> > (1) Transforming entangled multimodal features into explicit and semantically meaningful concept representations, corresponding to $F_{C^T}$, $F_{C^A}$, and $F_{C^V}$. Previous approaches struggle to extract features that truly correspond to multimodal cues (e.g. facial expressions, gestures, and vocal intonations) from raw multimodal inputs. While CBMs [6],[7] learn transformation matrices $W$ under KL-divergence supervision to obtain activation scores, they only capture cue presence rather than concrete manifestations (e.g., facial contours, body positions, or waveform patterns). ConMR advances by defining $W_{ij}$ as the relative contribution of the i-th concept along the j-th feature dimension, enabling element-wise multiplication with modality features to produce explicit concept features. Following [6], we sum explicit concept features to obtain concept scores supervised by $L_{KL}$, aligning feature activations with their corresponding concept labels. This approach separates sub-features within each concept’s semantic space from the entangled multimodal representations for intent reasoning. Ablation results show that replacing transformation matrix $W$ (w/o $W$) or removing explicit concept features (w/o $Z_{concept}$) significantly degrades MIntRec performance.
> >
> > (2) Introducing a concept-level multimodal reasoning path by modeling both concept-to-intent and inter-concept relations over explicit concept representations. In this reasoning process, key concepts from different modalities are jointly reasoned in a unified space. On the one hand, LLM guidance maps the relations between concept labels and intents onto their concrete manifestations through the $L_{MSE}$ objective. On the other hand, the joint activation patterns formed by all concept scores reveal higher-order interactions among concepts, enabling the modeling of complex inter-concept relations.
> > In summary, the novelty of our approach is substantial and fully justified.

---

> > > ### Author Response · Authors · 2025-11-21
> > > **Response to Reviewer XnHa (Part 3/3)**
> > >
> > > **A5** Thank you for raising the concern. We believe there may be a misunderstanding regarding our ablation study on LLM-derived priors, and we clarify the points as follows:
> > >
> > > (1) w/o $L_{KL}$ aims to demonstrate the importance of reliable concept activations for building robust representations. This setting indeed highlights the effectiveness of concept supervision derived from LLMs and the activation scores provided by the pretrained models.
> > >
> > > (2) w/o $W$ replaces the learnable transformation $W$ with linear layers but does not alter the LLM-derived supervision. The severe performance degradation observed under this setting is caused by removing the mechanism that transforms entangled multimodal features into concept representations, further validating the effectiveness of our approach.
> > >
> > > (3) w/o $L_{MSE}$ aims to show that LLM-based intent relevance score supervision is a critical component in concept-to-intent reasoning. The performance drop reflects the contribution of LLM-derived priors in guiding multimodal reasoning.
> > >
> > > (4) w/o $Z_{concept}$ is designed to verify the utility of the extracted concept representations in the concept-to-intent pathway. The substantial performance decline demonstrates the richness and expressiveness of these representations. Importantly, this ablation only removes the concept-to-intent pathway, while the inter-concept relation pathway remains supervised by $L_{MSE}$, ruling out the possibility that the observed drop is caused by the absence of LLM-derived priors.
> > >
> > > (5) w/o $Z_{inter}$ similarly validates the importance of concept activation scores in inter-concept reasoning. The observed performance drop is due to removing this pathway, not the absence of LLM-derived priors.
> > >
> > > Overall, these ablations comprehensively demonstrate the crucial contribution of each component in ConMR, rather than attributing performance solely to the absence of LLM-derived priors.
> > >
> > >
> > >
> > > **References**
> > >
> > > [1] Can Large Language Models Help Multimodal Language Analysis? MMLA: A Comprehensive Benchmark. Hanlei Zhang et al. NeurIPS 2025.
> > >
> > > [2] LLM-Guided Semantic Relational Reasoning for Multimodal Intent Recognition. Qianrui Zhou et al. EMNLP 2025.
> > >
> > > [3] LLM Reasoners: New Evaluation, Library, and Analysis of Step-by-Step Reasoning with Large Language Models. Hao, Shibo et al. COLM 2024.
> > >
> > > [4] A Survey on Hallucination in Large Language Models: Principles, Taxonomy, Challenges, and Open Questions. Lei Huang et al. ACM Transactions on Information Systems.
> > >
> > > [5] Evaluating Object Hallucination in Large Vision-Language Models. Yifan Li et al. EMNLP 2023.
> > >
> > > [6] CLIP-Dissect: Automatic Description of Neuron Representations in Deep Vision Networks. Tuomas Oikarinen et al. ICLR 2023.
> > >
> > > [7] Label-Free Concept Bottleneck Models. Tuomas Oikarinen et al. ICLR 2023.

---

### Author Response · Authors · 2025-11-22
**Summary of Our Revision**

We express our gratitude to all reviewers for their valuable time and constructive feedback. All concerns have been meticulously addressed, and we have made targeted revisions to the manuscript, with changes highlighted in blue in the updated version. The main modifications are summarized as follows.

**New Results in Revision**

1. In the main experiments, we conduct four additional baseline settings including LGSRR, Gemini-2.5 Zero-shot, Gemini-2.5 Few-shot and Gemini-2.5 CoT, which collectively show that ConMR consistently surpasses both MLLM pseudo-labeled baselines and MLLM zero/few-shot baselines. These results also substantiate our discussion of the challenges faced by existing methods (Page 6–7, Section 4, Table 1, **XnHa**).
2. We further include ablation studies on the concept selection strategy and the number of concepts, offering a deeper insight into the concept set of ConMR (Page 17-18, Appendix D, **HBfx**)
3. We additionally incorporate Qwen3-8B as the LLM backbone for ConMR, demonstrating that ConMR remains highly effective even under relatively low-resource settings. (Page 9, Section 5.4, Table 3, **dY2Z**).

**Revisions to the Text**

1. We clarify the limitations of the latest multimodal intent recognition method, LGSRR, showing that its shortcomings align closely with the challenges summarized in our work (Page 2, Section 1, **XnHa**).

---

### Author Response · Authors · 2025-12-01
**Summary of Our Response (Part 1/2)**

We express our gratitude to all reviewers for their valuable time and constructive feedback. All concerns have been meticulously addressed, with revisions highlighted in blue in the updated version. We are heartened by the positive evaluations of our work "This design improves interpretability, enabling the model to produce concept-level activations that correspond to human-understandable semantic units" (**XnHa**), "solidifying its value as a significant advancement towards trustworthy multimodal AI" (**dY2Z**) and "The attempt to transform feature-level patterns into the explicit concept-level paradigm is valuable" (**HBfx**).

Besides, we have also addressed the key issues and clarified several misunderstandings raised by the reviewers.

1. Advantages of ConMR over LLMs (**XnHa**). Our response clarifies that LLMs still face inherent limitations in multimodal intent understanding, even with strong prompting.  We support this claim with evidence from prior work as well as newly added baseline experiments, including LLM zero-shot, few-shot, CoT prompting, and the LLM-based LGSRR method. We additionally provide a focused analysis of the unresolved research challenges that persist for LLMs and demonstrate that our method achieves superior performance and interpretability compared with current MLLM approaches.
2. Novelty of ConMR (**XnHa**). We restate that the core novelty lies in transforming entangled multimodal features into explicit concept representations and introducing a concept-level reasoning path that models both concept-to-intent and inter-concept relations. We contrast this with prior CBM and fusion methods, which lack semantic granularity and fail to capture concept interactions. Formal definitions and targeted ablations further substantiate this clarification and eliminate ambiguity.
3. Concerns on Ablation Results (**XnHa**). We correct the misunderstanding by explaining that the ablation settings are designed to isolate the contributions of our concept-level reasoning modules rather than to eliminate LLM-derived supervision. The observed performance drops consistently arise from removing key reasoning components, demonstrating their indispensable role. This clarifies that the ablation results do not indicate dependence on LLM priors.
4. Performance in low-resource setting (**dY2Z**). We further evaluate ConMR by substituting the powerful Gemini-2.5 Flash with a significantly lighter LLM (Qwen3-8B). The results demonstrate that ConMR preserves strong performance and robust intent inference even under low-resource conditions, highlighting the effectiveness of its explicit concept representations and dual-path reasoning architecture.
5. Extended work on concept set refinement (**dY2Z**). We acknowledge that incorporating a feedback loop to iteratively refine the concept set is a promising future direction and propose a concrete refinement method that evaluates concept quality, removes low-impact or spurious concepts, and generates replacements via the concept pipeline to dynamically enhance reasoning accuracy, highlighting our method’s scalability and its potential as a robust multimodal concept-level reasoning paradigm.
6. Quality of LLM-generated concepts and ablations on concept selection (**HBfx**). We clarify that we have designed a comprehensive concept generation pipeline based on prior work to ensure high-quality concept set and relevance scores, which includes task-specific prompt design, filtering of redundant or intent-overlapping concepts, and submodular selection to maximize discriminability and coverage. Besides, our dual-path reasoning framework, combining concept-to-intent and inter-concept reasoning with weighting, ensures robustness against noisy concepts and biased relevance scores. Additional ablations on similarity filtering, submodular selection, and concept quantity confirm the effectiveness of each module and show that ConMR maintains strong, stable performance across concept variations.
7. Quality of LLM-generated concepts and ablations on concept selection (**HBfx**). We clarify that we have designed a comprehensive concept generation pipeline based on prior work to ensure high-quality concept set and relevance scores. Besides, our dual-path reasoning framework, combining concept-to-intent and inter-concept reasoning with weighting, ensures robustness against noisy concepts and biased relevance scores. Additional ablations on similarity filtering, submodular selection, and concept quantity confirm the effectiveness of each module and show that ConMR maintains strong, stable performance across concept variations.

---

> ### Author Response · Authors · 2025-12-01
> **Summary of Our Response (Part 2/2)**
>
> 8. Encoder choice and computational cost  (**HBfx**). We clarify that the encoders used are representative works in their respective domains and widely adopted in CBMs and multimodal intent recognition. Importantly, these encoders serve only for feature extraction and are not updated during training, which is standard practice in the field. As a result, the additional computational cost is minimal and generally accepted, and we do not consider it a significant concern.
> 9. Cross-modal concept interactions  (**HBfx**). We correct the misunderstanding that concept-level reasoning is limited to a single modality. In fact, ConMR performs unified reasoning over fine-grained semantic concepts extracted from multiple modalities. These concepts provide more fundamental and granular semantic units than modality-level features, capturing the key cues from which multimodal intent emerges and enabling more precise cross-modal reasoning than coarse-grained approaches.
> 10. Concerns on fixed concept set (**HBfx**). ConMR focuses on transforming entangled multimodal features into explicit concept representations and introducing a concept-level reasoning path that models both concept-to-intent and inter-concept relations. Accordingly, it follows the common CBM practice of using a pre-generated fixed concept set. As detailed in Appendix B, concept generation relies only on task background and labels, allowing generalization across benchmarks. For open-domain tasks, the concept set can be efficiently expanded, with new concepts incorporated via lightweight fine-tuning to handle novel tasks.

---

### Meta-Review · Area_Chair_vD8s · 2026-01-07

**Summary:**

The initial ratings are 4, 6, 2. This paper introduces ConMR of multimodal intent recognition that elevates reasoning from the feature level to the concept level. It is to leverage LLM-generated concepts as semantic anchors, learn explicit concept representations from multimodal features, and then perform structured reasoning over concept-intent and inter-concept relations.  Experiment results show good performance of the proposed method.

Strengths:
(1) This paper proposes a concept level reasoning approach for multimodal intent recognition, effectively bridging the gap between low-level features and high-level intents to enhance both performance and, crucially, model interpretability.
(2) The framework's robustness is confirmed by its consistent performance across different LLM, solidifying its value as a significant advancement towards trustworthy multimodal AI.

Weaknesses:
(1)Given that the proposed method already relies on LLMs for concept supervision and intent-relevance supervision, it is unclear why the authors do not directly employ LLMs for multimodal intent recognition (e.g., via zero-shot or fine-tuning approaches) instead of introducing additional concept reasoning layers.
(2) The novelty is overstated, and the motiation is not clear.
(3) This paper only compares with traditional fusion-based methods (MulT, MAG-BERT, MISA, etc.) and contrastive approaches (TCL-MAP, MVCL-DAF, etc.), but lacks LLM-based baselines

**Reviewer Concerns:**

Some concerns of Reviewer dY2Z were addressed by the rebuttal, and Some main concerns of  Reviewer HBfx and XnHa are still outstanding.

**Reviewer Scores:**

Most reviewers would not change the scores.

---

### Decision · Program_Chairs · 2026-01-26

Reject